# Divergent transcriptional and transforming properties of PAX3-FOXO1 and PAX7-FOXO1 paralogs

**Line Manceau**[1], **Julien Richard Albert**[1], **Pier-Luigi Lollini**[2], **Maxim V. C. Greenberg**[1], **Pascale Gilardi-Hebenstreit**[1]*, **Vanessa Ribes**[1]*

**1** Université Paris Cité, CNRS, Institut Jacques Monod, F-75013 Paris, France, **2** Department of Experimental, Diagnostic and Specialty Medicine (DIMES), Alma Mater Studiorum University of Bologna, Bologna, Italy

* pascale.gilardi@ijm.fr (PG-H); vanessa.ribes@ijm.fr (VR)

**Data Availability Statement:** The data will be mainly contained within the manuscript and/or Supporting Information files. In addition, raw and genome aligned sequencing data have been

## Abstract

The hallmarks of the alveolar subclass of rhabdomyosarcoma are chromosomal translocations that generate chimeric PAX3-FOXO1 or PAX7-FOXO1 transcription factors. Overexpression of either PAX-FOXO1s results in related cell transformation in animal models. Yet, in patients the two structural genetic aberrations they derived from are associated with distinct pathological manifestations. To assess the mechanisms underlying these differences, we generated isogenic fibroblast lines expressing either PAX-FOXO1 paralog. Mapping of their genomic recruitment using CUT&Tag revealed that the two chimeric proteins have distinct DNA binding preferences. In addition, PAX7-FOXO1 binding results in greater recruitment of the H3K27ac activation mark than PAX3-FOXO1 binding and is accompanied by greater transcriptional activation of neighbouring genes. These effects are associated with a PAX-FOXO1-specific alteration in the expression of genes regulating cell shape and the cell cycle. Consistently, PAX3-FOXO1 accentuates fibroblast cellular traits associated with contractility and surface adhesion and limits entry into S phase. In contrast, PAX7-FOXO1 drives cells to adopt an amoeboid shape, reduces entry into M phase, and causes increased DNA damage. Altogether, our results argue that the diversity of rhabdomyosarcoma manifestation arises, in part, from the divergence between the genomic occupancy and transcriptional activity of PAX3-FOXO1 and PAX7-FOXO1.

## Author summary

Rhabdomyosarcoma is a class of paediatric soft tissue cancers of genetic origin, but for which the causal links between genetic aberrations and tumour development remain to be deciphered. To answer this question, we focused on the products of two chromosomal translocations that generate the chimeric proteins PAX3-FOXO1 and PAX7-FOXO1. They are composed of the DNA-binding domains of the PAX3 or PAX7 proteins and a single portion of the FOXO1 protein. Several clinical parameters distinguish patients expressing PAX3-FOXO1 or PAX7-FOXO1 and we wondered if these differences could emanate from a different mode of action of the two chimeric proteins. Thus, we generated

deposited in the Gene Expression Omnibus (GEO) under accession number GSE180919 and is available at the following address: https://www.ncbi.nlm.nih.gov/geo/query/acc.cgi?acc=GSE180919.

**Funding:** VR is a staff scientist from the INSERM, PGH is a CNRS research director, MVCG is a CNRS staff scientist. LM has obtained a fellowship from University of Paris and her fourth year of PhD was supported by the Jacques Monod Institute and the Ligue contre le cancer (PREAC2020.LCC/MC). Work in the lab of VR was supported by the Ligue Nationale Contre le Cancer (PREAC2020.LCC/MC; PREAC2016.LCC; RS20/75-114). JRA is funded by a Fondation pour la Recherche Médicale Post-doc France Fellowship (#SPF202110014238) and work in the lab of MVCG was supported by the European Research Council (ERC-StG-2019 DyNAmecs). The funders had no role in study design, data collection and analysis, decision to publish, or preparation of the manuscript.

**Competing interests:** The authors declare that they have no known competing financial interests or personal relationships that could have appeared to influence the work reported in this paper.

inducible human fibroblast cell lines expressing one or the other protein. We analysed some molecular and cellular characteristics of these cells 48 hours after induction of PAX3-FOXO1 or PAX7-FOXO1. As it was previously known for PAX3-FOXO1, we showed that PAX7-FOXO1 binds genomic DNA on *cis*-regulatory regions and activates them. Surprisingly, PAX7-FOXO1 only partially shares the DNA binding sites of PAX3-FOXO1 and its activation potential is stronger than that of PAX3-FOXO1. Thus, PAX3-FOXO1 and PAX7-FOXO1 generate partially divergent transcriptomic signatures, which include genes encoding regulators of cell morphology and cell cycle, two key oncogenic processes. In agreement, our data revealed specificities in these two processes that are either PAX3-FOXO1 or PAX7-FOXO1 dependent. Overall our results demonstrate a differential mode of action between the two chimeric proteins that could in turn participate in the heterogeneity of rhabdomyosarcoma manifestation.

## Introduction

Fusion-positive rhabdomyosarcomas (FP-RMS) are one of the most metastatic and deleterious subgroups of paediatric soft tissue cancers [1]. Their emergence and development are closely associated with the activity of two paralogous fusion transcription factors (TFs), PAX3-FOXO1 and PAX7-FOXO1 [1] (S1A Fig). They are generated by the chromosomal translocations t (2;13)(q35;q14) or t(1;13)(p36;q14), which fuse the paired DNA-binding (PrD) and homeodomain (HD) domains of PAX3 or PAX7, respectively, to the transactivation domain of FOXO1. PAX3-FOXO1 and PAX7-FOXO1 positive tumour growths have broadly similar histological features, with clusters of round cells with sparse cytoplasm separated by fibrous septa [2]. In contrast, compared to patients expressing PAX7-FOXO1, those with the PAX3-FOXO1 translocation are more frequent, generally older, tend to harbour more metastases, and have poorer survival rates [3].

The clinical differences observed between patients harbouring the different translocations could arise from distinct and nonexclusive mechanisms. First, the specific anatomic positions of PAX3-FOXO1 and PAX7-FOXO1 tumours would argue for differing cells of origin (e.g. [4]). Second, genetic aberrations that accumulate following t(2;13) or t(1;13) translocations segregate PAX3-FOXO1 and PAX7-FOXO tumours, and may orient tumours towards distinctive states [5,6]. Both tumour types often undergo genome-wide duplication and carry focal chromosomal amplifications. While these amplifications extend the chromosomal segment carrying the PAX7-FOXO1 fusion, they increase the copy number of pro-proliferative genes in PAX3-FOXO1 tumours. Finally, each of the PAX-FOXO1 chimeric proteins could direct cells towards specific transcriptional states. Supporting this idea, PAX3-FOXO1 and PAX7-FOXO1 RMS differ in their DNA methylation profile and transcriptome [7–9], reflecting an altered *cis*-regulatory landscape [10].

The use of reporter transgenes demonstrated that the transactivation potential of PAX3-FOXO1 and PAX7-FOXO1 is much greater than that of the intact PAX3 and PAX7 TFs [11]. Furthermore, in several animal models, the forced expression of these two factors, when associated with pro-proliferative mutations, effectively generates growth masses reminiscent of patient-derived RMS tumours [12–15]. The molecular mechanisms underlying their tumorigenic transfection activity were mainly studied using PAX3-FOXO1 as a model [16–19]. PAX3-FOXO1 binds to non-coding *cis*-regulatory modules (CRMs) at DNA recognition motifs for its PrD domain or both its PrD and HD domains, but rarely exclusively on HD DNA binding motifs [16,17]. Furthermore, PAX3-FOXO1-bound CRMs are enriched in E-

BOX motifs recognized by bHLH TFs [16,17]. This is consistent with PAX3-FOXO1 acting with other TFs, including myogenic bHLH TFs in establishing central active CRMs in RMS cells [16, 20]. In addition, the bHLH TF MYOD1 has been shown to increase the transactivation potential of PAX7-FOXO1 as measured using transcriptional reporters [21], while both PAX-FOXO1 inhibit MYOD1-mediated myogenic differentiation [22]. Finally, characterization of the chromatin landscapes of PAX3-FOXO1-bound CRMs in FP-RMS cell lines, fibroblasts and myoblasts revealed that PAX3-FOXO1 act as a pioneer TF [16,23]. It can bind and open previously closed chromatin loci, which acquire the characteristics of potent transactivating enhancers (*e.g.* super enhancers). Importantly, the mechanisms by which the two fusion TFs regulate transcription have not yet been systematically compared [7,11]. Therefore, it is not known whether the recruitment and genomic trans-activity of PAX7-FOXO1 is akin to PAX3-FOXO1.

To gain insights into the role of the PAX-FOXO1 fusion TFs in tumorigenesis and heterogeneity, we directly compared their activities in a transgenic cellular model. By combining transcriptomic and chromatin profiling experiments with cell morphology and cell cycle assays, our results revealed a divergent transforming potential between the two PAX-FOXO1 proteins attributed to both differential use of their DNA binding domains and distinct transactivation potential.

## Results & discussion

### PAX3-FOXO1 and PAX7-FOXO1 harbour divergent genomic occupancy and transactivation potential

To compare the activities of the PAX3-FOXO1 and PAX7-FOXO1 fusion proteins, we sought to design a system in which their expression can be controlled and expressed at similar levels in an identical cellular context. Human foreskin fibroblasts (HFF) were engineered to express a copy of a FLAG-tagged version of these TFs in a doxycycline (DOX)-inducible manner from the *AAVS1* safeguard locus (S1B Fig). Three independent cell lines expressing PAX3-FOXO1 or PAX7-FOXO1 were generated (S1C Fig). All three PAX3-FOXO1 and PAX7-FOXO1 lines expressed similar fusion protein levels in bulk and at the single cell level 48h after exposure to DOX (S1C and S1E Fig).

Using these cell lines, we first mapped the genomic targets of PAX3-FOXO1 and PAX7-FOXO1 using Cleavage Under Targets and Tagmentation (CUT&Tag) [24] (Figs 1 and S2 and S3). We used two separate antibodies to detect the fusion TFs directed against either the N-terminal FLAG tag or the C-terminal FOXO1 domain. The latter could be used, as wild-type FOXO1 levels were low in control HFF compared to that of the transgenic PAX-FOXO1 proteins (S1C and S1D Fig). Data obtained with these two antibodies and in separate cell lines gave similar results, validating the approach (S2A Fig). In total, we selected 6000 *cis*-regulator modules (CRMs) with the highest PAX3-FOXO1 and/or PAX7-FOXO1 binding signals (Figs 1A, 1B and S2A and S1 Table). The majority of PAX-FOXO1-bound CRMs occupied intronic or intergenic regions (S2B Fig). These regions were mostly enriched for DNA motifs known to be recognized by PAX3 and PAX7 DNA-binding domains and poorly enriched for FOXO1 DNA binding motifs (Figs 1C, 1D and S2C and S2 Table). Hence, as shown for PAX3-FOXO1, PAX7-FOXO1 DNA sequence specific recognition is mainly mediated via its PAX7 DNA binding domains [16,17,23]. Finally, approximately 18% of these regions were previously identified as bound by PAX3-FOXO1 in an RMS cell line, some of which are in close proximity to iconic PAX3-FOXO1 target genes in RMS, including *MYOD1* and *PIPOX* (S2D and S2E Fig) [16]. Thus, the recruitment of PAX-FOXO1s to the genome is preserved in part across cellular contexts in which they act, and examination of PAX-FOXO1 activity on the fibroblast genome

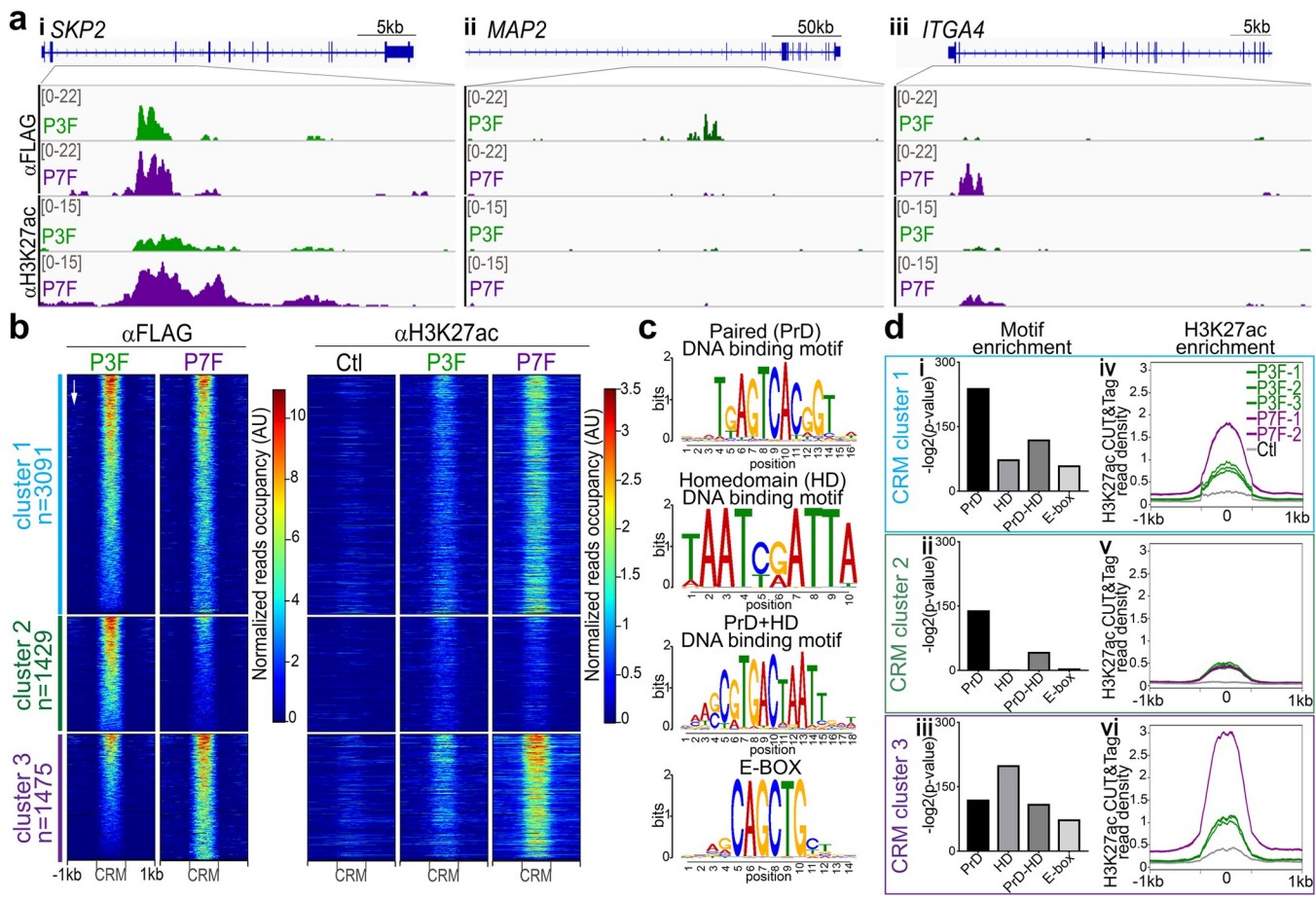

**Fig 1. PAX3-FOXO1 and PAX7-FOXO1 recruitment to the genome and impact on H3K27ac deposition. (a)** Examples of IGV tracks showing normalized FLAG and H3K27ac CUT&Tag reads distribution in cells expressing either PAX3-FOXO1 (P3F) or PAX7-FOXO1 (P7F) 48h post-DOX treatment. Scales in counts per million (CPM). **(b)** Heatmaps of normalized FLAG and H3K27ac CUT&Tag signals obtained in control (Ctl), PAX3-FOXO1 (P3F) and PAX7-FOXO1 (P7F) expressing HFF 48h post-DOX treatment in three *cis*-regulatory modules (CRMs) clusters. The ranking of the CRMs is the same on all heatmaps and follows the occupancy levels of PAX3-FOXO1 in descending order (arrow). Cluster 1 contains CRMs onto which the occupancy rate of PAX3-FOXO1 and PAX7-FOXO1 are similar. Cluster 2 CRMs are more bound by PAX3-FOXO1 than PAX7-FOXO1, and cluster 3 is the converse of cluster 2. **(c)** Logos of position weight matrices of DNA recognition motifs for the PAX paired (PrD) and/or homeodomain (HD) and by bHLH TFs (E-BOX). **(d)** Left panels: Enrichment for DNA binding motifs displayed in c in the CRMs belonging to the clusters defined in (b) (bars: -log2(*p*-value)). Right panels: Average profiles of normalized H3K27ac CUT&Tag signals at CRMs belonging to the clusters defined in (b) in the indicated control (Ctl), P3F or P7F cell lines. Note that the two P7F purple curves are superimposed.

is likely to shed some light on the contribution of both paralogs in establishing the molecular states of the RMS.

For approximately half of the total bound CRMs, the occupancy of PAX3-FOXO1 and PAX7-FOXO1 were comparable (Fig 1A–1I, cluster 1 in Fig 1B). In contrast, 1429 CRMs showed increased occupancy for PAX3-FOXO1 relative to PAX7-FOXO1 (Fig 1A-II, cluster 2 in Fig 1B) and 1475 CRMs the inverse (Fig 1A-III, cluster 3 in Fig 1B). Whereas PAX7-FOXO1 binding was barely detected in CRMs preferentially bound by PAX3-FOXO1, PAX3-FOXO1 recruitment was detected on most CRMs preferentially bound by PAX7-FOXO1 (Fig 1B). Thus, the two paralog proteins have divergent recruitment patterns in the genome, with PAX3-FOXO1 being more widespread than PAX7-FOXO1.

To test whether the observed differences in fusion protein occupancy may stem from a distinct use of their DNA-binding domains [25,26], we probed the different sets of CRMs for

several prototypical PAX3/7- and PAX3-FOXO1 PrD, PrD+HD, and HD DNA binding motifs (see Material and Methods; Fig 1C and S2 Table). All PAX-FOXO1-bound CRMs were enriched in PrD DNA binding motifs, to a lesser extent, in PrD+HD DNA binding motifs (Fig 1DI-III). The HD DNA recognition motifs were detected in CRMs bound by both PAX-FOXO1s (Fig 1DI). However, this motif was sparsely present in PAX3-FOXO1 CRMs (Fig 1DII) and, in contrast, most represented in PAX7-FOXO1 CRMs (Fig 1DIII). Overall, these results suggest that PAX3-FOXO1 uses its PrD and HD to bind DNA, whereas PAX7-FOXO1 preferentially uses its HD [25,26]. We then scanned PAX-FOXO1-related CRMs for E-BOX motifs recognized by class II bHLH TFs (Fig 1C and S2 Table). Apart from the PAX3-FOXO1-specific CRMs, this E-BOX was strongly represented in the other PAX-FOXO1s-bound CRMs, supporting a common interaction between bHLH TFs and both PAX-FOXO1s [16,20–22] (Fig 1DI-III).

We then assessed the chromatin status of PAX-FOXO1-linked CRMs in control and PAX-FOXO1s expressing HFF using CUT&Tag by mapping the genomic distribution of H3K27ac and H3K27me3, which are respectively hallmarks of transcriptionally active and repressed CRMs (Figs 1A, 1B, 1DIV-DVI and S3). We also analysed the distribution profile of the heterochromatin histone mark H3K9me3, which borders a fraction of PAX3-FOXO1 bound genomic regions in Rh4 RMS tumour cells [23]. Both PAX-FOXO1 proteins were recruited predominantly in CRMs which are in the control cell line (Ctl) in absence of fusion protein recruitment, unmarked by these three histone modifications with their levels very low compared to the regions of the genome more enriched for these marks (S3A–S3C Fig). The distribution profile and the levels of the heterochromatic H3K27me3 and H3K9me3 marks in the vicinity of these CRMs remained unchanged upon PAX-FOXO1 recruitment (S3A–S3C Fig). On a genome-wide scale, the occupancy rate of PAX-FOXO1s correlated better with *de novo* deposition of the H3K27ac mark than with the deposition of the heterochromatin marks (S3B Fig). Hence, PAX7-FOXO1, as PAX3-FOXO1, induces chromatin signatures of transcriptional activation [16]. Interestingly, H3K27ac levels showed a better correlation with PAX7-FOXO1 occupancy than with PAX3-FOXO1 occupancy (S3C Fig). Accordingly, the H3K27ac level on PAX-FOXO1-bound CRMs were generally higher in the presence of PAX7-FOXO1 than PAX3-FOXO1 (Figs 1A, 1B, 1DIV-DVI, S3A and S3CI). This was particularly visible on CRMs onto which both PAX3-FOXO1 and PAX7-FOXO1 could be recruited to (cluster 1 in Fig 1B). Thus, PAX7-FOXO1 ability to establish an active enhancer chromatin signature is greater than that of PAX3-FOXO1. This is reinforced by a principal component analysis of H3K27ac occupancy in the 6000 PAX-FOXO1 bound CRMs showing that ectopic PAX7-FOXO1 generates a chromatin state more distant from that of control cells than PAX3-FOXO1 (S3F Fig).

Overall, our results demonstrate that the two RMS-associated paralog fusion TFs exhibit a divergent mode of recruitment to the genome, likely due to the previously identified discrete affinities of the DNA-binding domains of PAX3 and PAX7 for specific DNA motifs [27,28]. Thus, the fusion between the PAX DNA-binding domains and the FOXO1 transactivation domain would have preserved the diversification that has occurred during evolution in the genome recruitment patterns of PAX3 and PAX7. Our data also support the idea that PAX-FOXO1 chimeras act as pioneer factors as previously shown [16,23]. Both are able to bind unmarked chromatin and have the capacity to promote its transcriptional activation. While the recruitment of PAX7-FOXO1 in most cases results in the deposition of the active enhancer mark H3K27ac, this was much less the case for PAX3-FOXO1. As previously observed [24], some PAX3-FOXO1 bound regions (mainly in cluster 2) did not engage towards transactivation. This lack of activation could be due to a lower enrichment in DNA motifs recognised by both the PrD and HD domains of the PAX-FOXO1 or by the bHLH TFs. Supporting the former, it has been shown for the intact Pax7 protein that recruitment to a PrD-like motif is less

likely to result in transcriptional activation than recruitment to motifs recognised by both PrD and HD. Finally, it is notable that the repertoire of DNA motifs in the PAX-FOXO1-bound regions we identified in HFF is much less enriched in juxtaposed PrD and HD motifs than the PAX3-FOXO1-bound regions in tumour cells [26]. This could be due to a temporal dynamic in the recruitment mode of the PAX-FOXO1 factors. Indeed, it has been shown that some pioneer factors begin to be recruited to DNA first on incomplete DNA recognition motifs and then on more complete motifs [27]. Supporting this idea, in myoblasts, PAX3-FOXO1 is less recruited to PrD+HD sites after 8 hours than after 24 hours [23]. Finally, in the tumour context, the specificities in the transcriptional activity of PAX3-FOXO1 and PAX7-FOXO1 that we have revealed in fibroblasts can be expected to be amplified or diminished by other parameters, such as the presence of several isoforms with variable transcriptional potential [28], variations in the expression levels of PAX-FOXO1s [29], or the cellular context (cells of origin and history).

## PAX3-FOXO1 and PAX7-FOXO1 induce unique transcriptomic landscapes

To investigate the phenotypic consequences of the divergent activity of PAX3-FOXO1 and PAX7-FOXO1 on chromatin, we first compared the RNAseq-based transcriptome of HFF expressing either chimeric TF or control HFF (Figs 2A and S4A and S3 Table). Mirroring the differential H3K27ac landscapes, the transcriptome of PAX7-FOXO1 cells diverged more from controls than the transcriptome of PAX3-FOXO1 cells (S4AI and S4AII Fig). We identified the genes whose expression varied the most between samples, which likely underlay the segregation of the transcriptome, and used K-means clustering heatmaps to highlight their behaviour in the samples (Figs 2A and S4AIII). PAX3-FOXO1 and PAX7-FOXO1 decreased the expression of the same subset of genes (cluster A in Fig 2A and S3 Table). This subset was weakly enriched in genes in the vicinity of the identified PAX-FOXO1 recruitment sites, and was therefore unlikely to be directly regulated by PAX-FOXO1s (Fig 2BI). In contrast, the three clusters of genes upregulated by PAX3-FOXO1 and/or PAX7-FOXO1 were significantly enriched in genes nearby PAX-FOXO1s bound CRMs (clusters B to D in Fig 2A and 2BII-IV and S4AIII Fig). In addition, genes more induced by PAX7-FOXO1 than by PAX3-FOXO1 were mainly enriched in CRMs preferentially bound by PAX7-FOXO1 (cluster D in Fig 2A and 2BIV and S4AII Fig). Conversely, genes on which PAX3-FOXO1 had a greater impact than PAX7-FOXO1 were predominantly enriched in CRMs bound by PAX3-FOXO1 (cluster C in Fig 2A, 2BIII and S4AIII Fig). Thus, differences in PAX-FOXO1 recruitment are responsible for differential gene activation. Consistent with this observation, genes located near CRMs that can be bound by both PAX-FOXO1s were generally induced in their presence, with higher levels in the presence of PAX7-FOXO1 than PAX3-FOXO1 (S4AV Fig, left panel). While genes close to the CRMs preferentially bound by one of the two PAX-FOXO1s displayed higher expression levels in cells expressing the PAX-FOXO1 that binds these CRMs, again with a higher activation for genes near CRMs bound by PAX7-FOXO1 than PAX3-FOXO1 (S4AV Fig, middle and right panels). Importantly, this further supports a greater trans-activation potential of PAX7-FOXO1 than PAX3-FOXO1. We next compared the transcriptomes of cells transiently expressing murine *Pax3*, *Pax7*, or *PAX3-FOXO1* or *PAX7-FOXO1* (S4B Fig). Genes induced by both PAX3-FOXO1 and PAX7-FOXO1 were poorly upregulated by Pax3 and Pax7. In contrast, genes that were specifically induced by one of the PAX-FOXO1 paralogs were also induced by its wild-type PAX variant and were not induced by the PAX paralog variant (Figs 2C and S4B). Therefore, the induction of specific gene cohorts by one of the PAX-FOXO1 chimeras is a property that emanates from its PAX moiety.

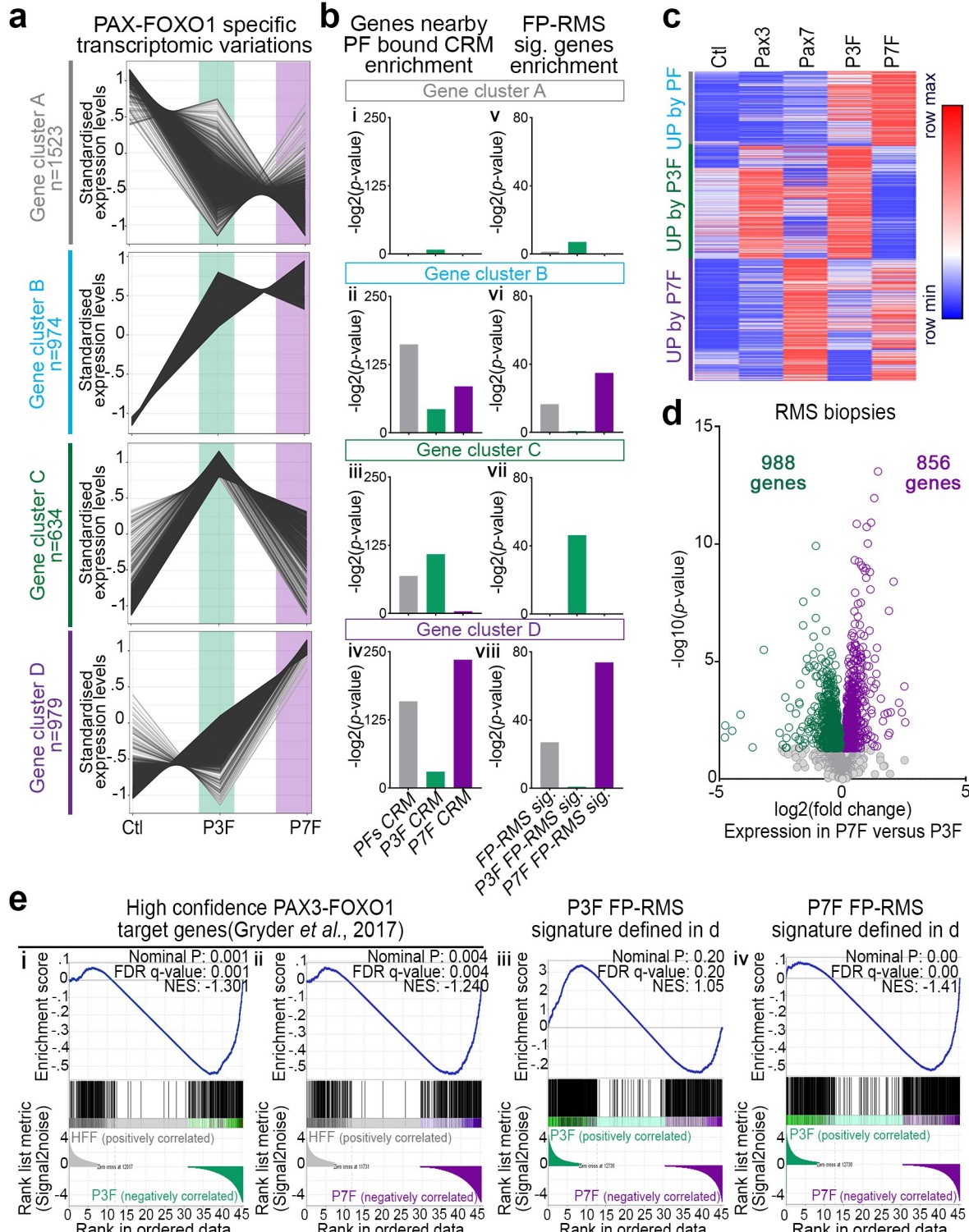

**Fig 2. Characterization of PAX3-FOXO1 and PAX7-FOXO1 specific transcriptomic signatures. (a)** Standardized and normalized expression levels of genes assayed by RNAseq and belonging to the 4 K-means clusters defined in S4AIII Fig and showing significant variations in their expression between control, PAX3-FOXO1 (P3F) and PAX7-FOXO1 (P7F) expressing cells. **(b)** Left panels: Enrichment for genes in the vicinity of *cis*-regulatory modules (CRMs) defined in Fig 1B (PFs CRMs, P3F CRMs, P7F CRMs are respectively clusters 1,2,3 in Fig 1B) in the gene clusters defined in (a). Right panels: Enrichment for genes associated with FP-RMS[11] (FP-RMS sig.) and genes associated with PAX3-FOXO1 (P3F FP-RMS sig.) and PAX7-FOXO1 (P7F FP-RMS sig.) in RMS biopsies in the gene clusters defined in

(a). Bars: -log2(*p*-value). **(c)** Heatmap of K-means clustered genes that displayed higher expression in P3F and/or P7F transfected cells compared to control cells. Fold changes across Pax3, Pax7, P3F and P7F samples are colour coded in blue (lower level) and in red (higher level). **(d)** Volcano plot comparing gene expression levels assayed by microarray between PAX3-FOXO1 (P3F) and PAX7-FOXO1 (P7F) biopsies, with statistical significance (-log10(*p*-value)) on the y-axis versus the magnitude of change (log2(fold change in PAX7-FOXO1 samples compared to PAX3-FOXO1)) on the x-axis. Grey dots: not significantly differentially expressed genes, green dots: significantly upregulated genes in PAX3-FOXO1 samples, purple dots: significantly upregulated genes in PAX7-FOXO1 samples. **(e)** GSEA plots showing the enrichment for genes associated to PAX3-FOXO1 activity in RMS [16] or to PAX3-FOXO1 and PAX7-FOXO1 activity in RMS (cf. d) in differentially expressed genes between control (Ctl) HFF and HFF expressing either PAX-FOXO1s (PF) or PAX3-FOXO1 (P3F) or PAX7-FOXO1 (P7F) or between P3F and P7F samples. NES: normalized enrichment score; FDR: false discovery rate.

Importantly, the genes induced by PAX3-FOXO1 and by PAX7-FOXO1 in HFF were significantly enriched for gene signature characteristics of fusion positive RMS (Figs 2EI-II and S5A). Thus, we next wondered whether they could explain fusion subtype-dependent variation in transcriptomic status of RMS tumours [8,9,30]. We first evaluated the specific gene signatures of the PAX-FOXO1 paralog in RMS using previously published microarray data on 99 PAX3-FOXO1 and 34 PAX7-FOXO1 biopsy samples [15] (Fig 2D and S4 Table). We identified 988 genes more enriched in PAX3-FOXO1 biopsies than in PAX7-FOXO1 biopsies and 856 genes specifically enriched in PAX7-FOXO1 tumours. Functional annotation of these genes highlighted that the PAX3-FOXO1-specific signature was enriched in regulators of cell cycle, cell migration, and metabolism (S5B Fig and S5 Table). The genes associated with PAX7-FOXO1, on the other hand, encoded regulators of embryonic lineage differentiation as well as genes involved in cytoskeletal remodelling (S5C Fig and S5 Table). Thus, paralog-specific transcriptional states could confer particular cellular traits to the cell transformation. Importantly, enrichment analyses indicated that PAX3-FOXO1 and PAX7-FOXO1 specific gene signatures in RMS and HFF were significantly overlapping (Fig 2BV-VIII and 2EIII-IV). Hence, PAX-FOXO1-dependent genetic signatures established in the cell of origin would be preserved despite the accumulation of genetic aberrations during cell transformation [5,6] and could provide specific tumorigenic features [3]. Consistent with this idea, the function of genes differentially expressed between PAX3-FOXO1- and PAX7-FOXO1-expressing HFF was reminiscent of those differentially expressed between (t2;t13)- and (t1;t13)-carrying RMS cells (S5D and S5E Fig and S5 Table).

## PAX7-FOXO1 induces greater alterations of cell architecture and cell cycle than PAX3-FOXO1

The enrichment for genes encoding cell cycle and cell shape regulators in the specific signatures of PAX-FOXO1s (S5 and S6 Figs), led us to analyse their functional impact on these two cellular parameters.

First, transcriptional analyses revealed that control fibroblasts and those expressing PAX3-FOXO1 and PAX7-FOXO1 differentially expressed specific members of the integrin, cadherin, semaphorin, small GTPase, guanine nucleotide exchange factor (GEF), GTPase activating protein (GAP), or actin binding and processing protein families (S6A Fig and S3 Table). Hence, they likely possess their mechanism for regulating acto-myosin network dynamics and cell adhesions (S6A Fig). Accordingly, cell morphology in these samples was distinct. This was revealed by monitoring the functional architecture of the cytoskeleton using phalloidin-labeled F-actin and focal adhesions immunostained with anti-Paxillin (Figs 3A and S7I-V). While control and PAX3-FOXO1-expressing cells predominantly exhibited a spindle or triangular shape, more than 35% of PAX7-FOXO1 cells became rounded (Figs 3AI-IX and S7IV). Stress fibres terminated by focal adhesions were barely visible in PAX7-FOXO1 cells (Figs 3AIV and S7IV-V); instead, these cells displayed filopodia-like F-actin microspikes (arrowheads in Fig 3AIV').

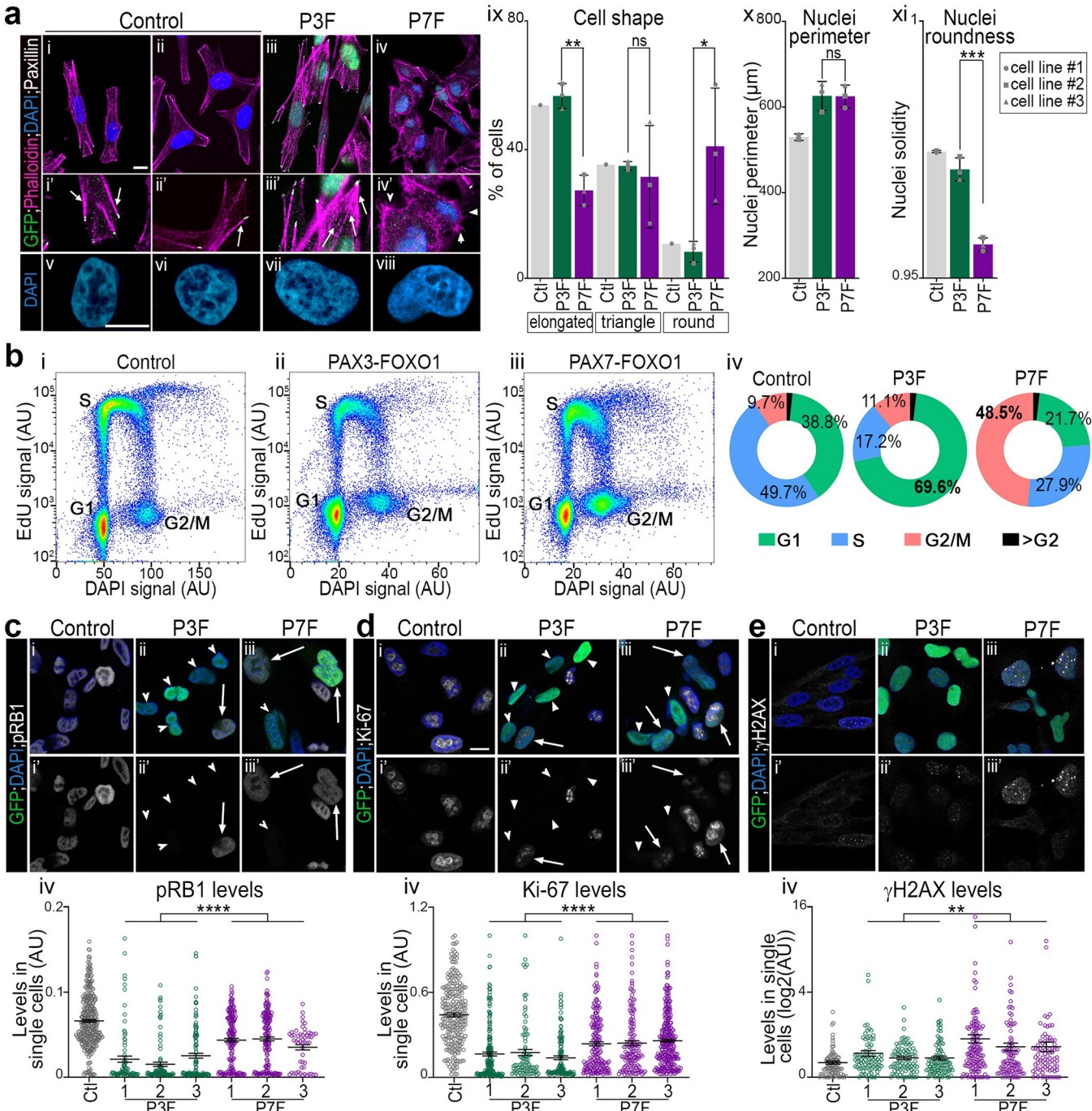

**Fig 3. PAX3-FOXO1 & PAX7-FOXO1 activities differentially impact the shape of cells and their cycling behaviour. (a)(i-viii)** Immunodetection of Paxillin and GFP, phalloidin based F-actin labelling and DAPI staining on the indicated HFF cell lines treated for 48h with DOX. **(ix-xi)** Proportion of cells harbouring the indicated cell morphology (ix), quantification of nuclei perimeter (x) and solidity (xi) in the indicated cell lines treated with DOX for 48h (bars: mean ± s.d.; dots: mean value in independent cell lines; Mann-Whitney U test $p$-value: *: $p < 0.05$, **: $p < 0.01$, ***: $p < 0.001$, ns: $p > 0.05$). **(b)(i-iii)** FACS plots displaying EdU levels and DNA content (DAPI levels) in control cells and in the GFP+ population of a PAX3-FOXO1 and PAX7-FOXO1 cell line treated for 48h with DOX. **(iv)** Percentage of cells in the indicated cell cycle phase established from plots in i (mean over three experiments and three independent lines). **(c-e)(i-iii')** Immunodetection of pRB1, Ki-67, γH2AX, 53BP1 and GFP and DAPI staining on the indicated HFF cell lines treated for 48h with DOX. **(c-e)(iv)** Levels of expression of pRB1, Ki-67 and γH2AX in control, GFP+ PAX3-FOXO1 and GFP+ PAX7-FOXO1 cells treated for 48h with DOX (dots: cell values, bars: mean ± s.e.m., two-way-ANOVA $p$-values evaluating the similarities between PAX3-FOXO1 and PAX7-FOX1 cells lines: ****: $p < 0.0001$).

In contrast, in cells expressing PAX3-FOXO1, stress fibers were greater in number and thicker than in control cells (Figs 3AIII' and S7II-III). This indicates that each PAX-FOXO1 paralog differentially alters actin network dynamics, with PAX3-FOXO1 favouring contractile actomyosin bundles. Further demonstrating the PAX-FOXO1 paralog specific cell shape dynamics were variations in the nucleus shape, one of the key organelles of cell proprioception [31] (Fig 3AV-VIII and X-XI). Expression of both PAX-FOXO1 chimeric proteins is associated with the enlargement of HFF nuclei (Fig 3A-X). Yet, consistent with the loss of actin-based tension in PAX7-FOXO1 cells, their nuclei lost their roundness and adopted a bean or multilobed shape (Fig 3A–3C and 3XI). Overall, these results indicate that PAX3-FOXO1 enhances cellular features related to cell contractility and surface adhesions already present in HFF. In contrast, PAX7-FOXO1 orients cells towards a distinctive amoeboid-like morphological state. With such clear differences in the morphological characteristics of HFF expressing PAX3-FOXO1 and PAX7-FOXO1, it is tempting to hypothesise that the two factors may confer on the cells distinct capabilities in terms of tissue invasions [32].

To assess the impact of PAX-FOXO1s on the cell cycle, we labelled replicating cells with a 1h30 pulse of thymidine analogue EdU prior to harvest. We then used flow cytometry to quantify DNA content and replication using DAPI and EdU levels (Fig 3B). Both PAX-FOXO1s altered the distribution of cells across cell cycle phases (Fig 3B). The number of cells in the replication (S) phase was reduced in the presence of both chimeras, but this reduction was much greater in the presence of PAX3-FOXO1 than PAX7-FOXO1 (Fig 3B). Consistently, CYCLIN-CDK-dependent phosphorylation levels of Rb1 and the cell proliferation marker Ki-67 (MKI67) were further reduced in PAX3-FOXO1 cells compared to PAX7-FOXO1 cells (Figs 3C, 3D and S7VI-XV). This is also in agreement with the decreased expression of core cell cycle genes and the upregulation of cell cycle inhibitors by both PAX-FOXO1s (S6B Fig). Interestingly, whereas the G1 cell population was dominant in PAX3-FOXO1-expressing HFF, PAX7-FOXO1 expression was associated with an increased percentage of cells in G2 (Fig 3BII-IV). This observation was reinforced by the enrichment of PAX7-FOXO1 molecular signature for genes encoding regulators of the G2 to M transition, whereas PAX3-FOXO1 specifically induced inhibitors of the G1 to S transition (S6B Fig and S3 Table). PAX3-FOXO1 expression levels and transcriptional potential in several PAX3-FOXO1 positive RMS models have been shown to be potentiated during the G2 phase [33]. We thus verified that the pausing in G1 of PAX3-FOXO1+ cells and in G2 of PAX7-FOXO1+ does not alter either the expression of PAX-FOXO1 under the control of the DOX promoter or the PAX-FOXO1-specific response of the target genes (S8 Fig). Finally, we tested whether PAX-FOXO1s-mediated cell cycle dysregulation could be correlated with DNA damage. Indeed, such lesions are likely to block cell cycle progression [34]. Immuno-labelling of 53BP1+ and γH2AX+ DNA double-strand break foci showed that PAX7-FOXO1 was more prone to cause genomic instabilities than PAX3-FOXO1 (Figs 3E and S7XVI-XX).

The distinct effects of the different PAX-FOXO1 fusion proteins on cell cycle and DNA damage raise the question of the mechanisms by which the two PAX-FOXO1s bypass their cell cycle block. In the case of PAX3-FOXO1, the appearance of genetic aberrations increasing the expression of MYCN and CDK4 [6], two regulators of the G1 to S transition, presents an efficient and necessary means for the generation of tumour growths [12–15]. In addition, the respective PAX-FOXO1 activity could also promote cell cycle continuation [33]. Indeed, in a mouse model of RMS, PAX3-FOXO1 activity has been shown to facilitate checkpoint adaptation and is, for example, required for the entry into mitosis of DNA-damaged cells. The effect of PAX7-FOXO1 expression on the cell cycle remains unexplored. Nevertheless, the more pronounced impact of PAX7-FOXO1 on the transcriptional landscape of cells and their morphology, but also on features such as DNA breaks that are known to ultimately lead to cell death

[35], leads us to propose that in the presence of PAX7-FOXO1, cells will be faced with greater obstacles to tumour formation than in the presence of PAX3-FOXO1. Although this hypothesis remains to be tested, it offers a plausible explanation for the lower occurrence of PAX7-FOXO1 tumours compared to PAX3-FOXO1 tumours. The extinction of PAX7-FOXO1 expression, despite amplifications of the genomic region carrying the fusion, supports the idea that cells are less tolerant to PAX7-FOXO1 than to PAX3-FOXO1 [29].

Overall, our study indicates that within 48 hours the paralogous oncogenic fusion factors PAX7-FOXO1 and PAX3-FOXO1 each impose specific molecular and cellular characteristics on fibroblasts. Among the underlying mechanisms, recruitment to distinct genome sites and divergence in their ability to transactivate these sites may be of particular importance. Finally, the presence of certain molecular characteristics associated with a given PAX-FOXO1 in both fibroblasts and tumours suggests that their specific mode of action remains preserved in different contexts of the cells and/or the tumorigenesis process. Yet, the variations in the response of several embryonic cell lineages to PAX3-FOXO1 [14] and the large spectrum of cells that could participate to the emergence of the rhabdomyosarcoma, including connective tissue cells and myoblasts [12,36] urge to understand how cell context could modulate in the modes of action of the two fusion proteins. It will be also important to assess in the future how the activity of these paralogous factors, but also of other fusions between the DNA-binding domains of PAX and transcriptional regulators other than FOXO1, such as NCOA1 [37], modulates the response of cells in models recapitulating the physiology of tissue interactions, in order to determine whether their differential transcriptional activity is able to generate diversity in tumour manifestation.

## Material and methods

### Plasmid constructs

*2xFLAG-PAX3-FOXO1-IRES-NLS-GFP* and *2xFLAG-PAX7-FOXO1-IRES-NLS-GFP* cassettes were PCR amplified from previously described constructs [15,38] using the primers P1,P2 (see below), respectively and cloned in *pAAVS1-PDi-CRISPRn* plasmid [39] between PacI and NotI restriction sites using Gibson assembly technology. *pAAVS1-PDi-CRISPRn* was a gift from B. Conklin (http://n2t.net/addgene:73500; RRID:Addgene_73500). In these constructs, PAX3-FOXO1 sequences encode for the Q+PAX3-FOXO1 isoform and PAX7-FOXO1 sequences encode for the Q+GL+ PAX7-FOXO1 isoform [28]. All constructs were verified by DNA sequencing (MWG, Eurofins).

P1: CTACCCTCGTAAACTTAAGGTTAATGTCTCATCATTTTGGCAAAGAATTG
P2: CAAGTTGGGGGTGGGCGATCGATTGCGTCGACGGTGCAGGT

### Cell lines description and maintenance

Control and engineered Human Foreskin Fibroblasts (HFF) [40] were grown as described [41]. Doxycycline (DOX) was used at a concentration of 1μg/ml.

### Generation of stable cell lines

To create DOX-inducible PAX-FOXO1 HFF lines, 2-3x10$^5$ plated cells at a passage 7 were transfected with *pAAVS1-PDi-CRISPRn* containing a *2xFLAG-PAX-FOXO1-IRES-NLS-GFP* cassette and the sgRNAs *gRNA_AAVS1-T1* (http://n2t.net/addgene:41817; RRID: Addgene_41817) and *gRNA_AAVS1-T2* (http://n2t.net/addgene:41818; RRID: Addgene_41818) (Gifts from G. Church [42]) using Lipofectamine 2000 (Life technologies, Carlsbad, CA, USA) and the manufacturer's recommendations. 24h later, puromycin (2μg/ml)

was added to the medium. After 2 days, cells were split. 10 days later, resistant cells were treated with 1μg/ml of DOX during 48h and then GFP$^+$ cells were sorted with FACSAria Fusion (BD-Biosciences). The sorted cells were then grown and amplified without DOX. Three independent PAX3-FOXO1 and three PAX7-FOXO1 lines were obtained, referred to here as P3F-1 to 3 and P7F-1 to 3.

## CUT&Tag, sequencing and analyses

CUT&Tag was performed as in Kaya-Okur *et al.* [9], following the procedure described in "Bench top CUT&Tag V.2" available on protocols.io. We used 100 000 cells per sample and the anti-FLAG (Merck F3165), anti-FOXO1 (Cell Signaling #2880), anti-H3K27ac (Diagenode C15410196), anti-H3K27me3 (Diagenode C15410195) and H3K9me3 (Diagenode C15410193) primary antibodies and anti-rabbit IgG (ABIN6923140) or anti-mouse IgG (ABIN6923141) secondary antibodies. For FLAG, FOXO1 and H3K27ac libraries, we used pA-TN5 from Tebu-Bio (Epicypher #15–1017). For H3K27me3 and H3K9me3 libraries, we used a homemade purified pA-TN5 protein following a previously published protocol [10]. Libraries were analysed using 4200 TapeStation system (Agilent) and sequenced 150pb (paired-end) with HiSeq X-ten (BGI). The quality of paired-end read sequences was assessed using FastQC (v0.11.9). Low quality nucleotides and Illumina adapter sequences were removed using Trimmomatic (v0.39) [11] and PCR duplicate reads were removed using BBmap clumpify (v38.87) [12]. Filtered reads were aligned to the hg19 reference genome using Bowtie2 (v2.4.1) [13, 14] as described previously [15]. Genome-wide coverage tracks (bigwigs) were generated using deeptools (v3.3.0) [43] and normalised by library size (Counts Per Million, CPM) with default parameters. Peaks were called using SEACR (v1.3) [16] with parameters:—relaxed—threshold 0.000001 and manually curated. Reads coverage on defined CRMs was calculated using VisRseq (v0.9.2) [44] and normalised by library size. The top 6000 PAX-FOXO1s bound peaks identified with anti-FLAG and/or anti-FOXO1 were kept. Differential analysis on these 6000 peaks was performed using Deseq2 (v1.30.1): 3091 were bound by both PAX-FOXO1s (cluster 1), 1429 were statistically enriched (*p*-adj<0,01) for more PAX3-FOXO1 (cluster 2) and 1475 were statistically enriched for more PAX7-FOXO1 (cluster 3). Histone marks signal peaks common to all cell lines analysed (cf. S3 Fig) were identified using bedtools multiple intersect on the Galaxy web platform. Heatmaps and average plots were generated using deeptools (v3.3.0) [17] computeMatrix followed by plotHeatmap or plotProfile. PCA plots were generated with deeptools MultiBamSummary and visualized with plotPCA with the—transpose parameter enabled. The nearest gene to each peak was determined using bedtools closest (v2.27.0). Heatmaps of relative expression levels were generated using Morpheus (https://software.broadinstitute.org/morpheus/). Correlograms were generated with the Morpheus Similarity Matrix tool using Spearman rank correlations of RPKM values calculated over 10kb genomic bins with VisRseq [44].

## Motifs analyses

Position weight matrices (PWMs) for PAX-FOXO1s binding sites were taken from JASPAR database [45] or created *de novo* from Pax3, Pax7 or PAX3-FOXO1 ChIP-Sequencing data [16,26] using RSAT [46,47]. Pax3, Pax7 and PAX3-FOXO1 peaks location data were retrieved for the top 500 ChIPseq peaks and 100bp sequences either side of their centre were fetched using Galaxy (usegalaxy.org). In total, 21 PWM were identified and used to calculate the relative enrichment of each of PAX as well as E-BOX and FOXO1 related DNA motifs in CUT&-Tag data using AME [48] from MEME-suite (S2 Table). The logos of the 5 most recurring

motifs were generated from PWM using https://ccg.epfl.ch/pwmtools/pwmscan.php (Figs 1C and S2C). Results for these motifs are shown in the left panels of Fig 1D and in S2CII Fig.

## RNA extraction, sequencing and analyses

RNAs were extracted using the NucleoSpin RNA kit (Macherey-Nagel) following manufacturer's instructions, elution was done with RNase-free water and concentrations were measured with DeNovixDS-11-FX series. RNAs were sent to Fasteris (Switzerland) where libraries were prepared and their quality checked using a Bioanalyzer Pico chip. 0.2–0.5μg was then either sequenced on Paired-end reads 100 base runs, on an Illumina Novaseq instrument (Stable HFF cell lines) or on paired-end reads 75 base runs on a HiSeq instrument (Transient transfection in HFF). After demultiplexing, reads were trimmed and mapped on the hg19 (Stable HFF cell lines) or hg38 (Transient transfection in HFF) human genome (GSE180919).

## Transcriptomic analyses

Pairwise differential analysis of RNAseq data was performed using Cufflinks (Fasteris). Genes with a *q*-value <0.05 after Benjamini-Hochberg correction for multiple-testing were considered to be significant. Hierarchical clustering was performed with one minus Pearson correlation using Morpheus. MDS was performed using the Cummerbund package. The normalized FP-RMS biopsies transcriptomes evaluated using microarrays have been previously described [15]. Genes differentially expressed between the 99 PAX3-FOXO1 and 34 PAX7-FOXO1-positive RMS were evaluated using paired *t*-test and the cut-off was set for a *p*-value <0.05. Biological pathways GO terms enrichment analyses were performed with EnrichR (http://amp.pharm.mssm.edu/Enrichr/). Homemade enrichment analyses between data sets were performed using hypergeometric tests (phyper function on R). A homemade lists of genes coding for regulators of cell morphology, cell adhesion and motility and of cell cycle linked genes were generated based on literature reviewing and were used to picture the effects of the PAX-FOXO1s onto these biological processes (S6 Table and S6 Fig).

## Immunohistochemistry

Cells were plated in 24-well culture plates containing coverslips and allowed to adhere at 37˚C O/N with DOX. After 48h, cells were fixed with 4% paraformaldehyde (PFA) for 8 minutes and processed for immunostaining [49]. The following antibodies diluted in PBS, 1% BSA, 0.1% triton were used. *Primary antibodies*: rabbit anti-53BP1 (Novus, NB100-304SS; 1/1000), chicken anti-GFP (Abcam, ab13970; 1/1000), rabbit anti-FOXO1 (Cell signaling, #2880; 1/1000), mouse anti-γH2AX (Millipore, 05–636; 1/200), rabbit anti-Ki-67 (Abcam, ab15580; 1/1000), mouse anti-PAX3/7-HD (clone DP312 Nipam Patel's lab; 1/100), mouse anti-Paxillin (Santa-Cruz, sc390738; 1/100), rabbit anti-phosphorylated RB1 (pRB1)(R&D systems, MAB6495, 1/500). *Secondary antibodies*: donkey against mouse and rabbit IgG or chicken IgY coupled to Alexa Fluorophores A488, A568 or A647 (Thermo Fisher Scientific) were diluted 1:500 and used with DAPI (500ng/ml, Sigma, 28718-90-3) or Hoechst (1/10000, Thermo Fisher Scientific, H3570). F-actin was marked by adding to the secondary mix Rhodamine Phalloidin (Cytoskeleton, PHDR1; 1/500). Immunofluorescence microscopy was carried out using a Leica TCS SP5 confocal microscope, a Leica DMR Upright or Zeiss LSM980 spectral Airyscan 2. All the images were processed with Image J v.1.43g image analysis software (NIH) and Photoshop 13.0 software (Adobe Systems, San Jose, CA, USA). All quantifications were performed using CellProfiler v3.1.5 (http://www.cellprofiler.org/). Statistical analyses and graphical representation were performed using a Mann-Whitney U or t-tests or two-way ANOVA test in GraphPad Prism and *p*-values are given in figure legends.

## Western blots

Cells were lysed in a buffer composed of 10mM Tris-Cl pH7.5, 5mM EDTA, 150mM NaCl, 30mM Sodium pyrophosphate, 50mM Sodium fluoride, 10% glycerol, 1% NP40 and Complete Protease Inhibitor Cocktail (Sigma-Aldrich). 10μg of cellular extracts were separated on 10% SDS-PAGE under reducing conditions and transferred onto polyvinylidene difluoride membranes with iBlotTM2 Gel Transfer Device (Thermo Fisher Scientific). After protein transfer, the membranes were blocked for 1 hour in TBST buffer containing 10% milk followed by incubation with primary antibodies (rabbit anti-FOXO1, #2880, Cell Signaling; 1/1000; rabbit anti-βACTIN, #4967, Cell Signaling; 1/1000) overnight at 4°C. After washes, membranes were incubated for 2 hours with secondary antibody (Goat anti-Rabbit Secondary, Alexa Fluor 700, Invitrogen; 1/10000). Fluorescent bands were visualized on LI-COR Odyssey and their intensity quantified and normalized using Image J v.1.53i image analysis software (NIH). Statistical analyses and graphical representation were performed using two-way ANOVA test in Graph-Pad Prism and all the *p*-values are given in figure legends.

## Cell cycle analyses

10μM EdU were added to cells 48h post-DOX induction. After 1h30, they were harvested and washed twice with PBS. Cells were incubated for 30 min with chicken anti-GFP (Abcam, ab13970; 1/2000) and then 30 min with Alexa Fluorophores A488 coupled secondary antibodies (Thermo Fisher Scientific; 1/2000). After three PBS washes, click-it reactions were performed using the Click-iT Plus EdU Alexa Fluor 647 Flow Cytometry Kit (Thermo Fischer Scientific) according to the manufacturer's recommendations. Cells were counterstained with DAPI for 30 min. Sample measurements were done on a BD FACSAria Fusion flow cytometer (BD Biosciences) and with FACSDiva Software (v8.0.1, BD Biosciences). Alexa Fluor 488 signal was measured by 488nm laser excitation with a 530/30 bandpass filter, Alexa Fluor 647 by 633nm laser with a 670/30 nm bandpass filter and DAPI by 405nm laser with a 450/40 nm bandpass filter. Data analysis was performed with FlowJo v10.7.1 Software (BD Biosciences). Cells were identified on a Side Scatter (SSC) vs Forward Scatter (FSC) dot plot and cell debris and aggregates were excluded from analysis based on FSC signals. Cell cycle was studied on three independent cell lines, in three independent experiments.

## RT-quantitative real-time-PCR on FAC sorted G1 and G2 cells

Three sets of control HFF and the three independent PAX3-FOXO1 (P3F-1 to 3) and PAX7-FOXO1 (P7F-1 to 3) cells were treated with DOX for 48h and with Hoechst 33342 (Thermo Fisher Scientific; concentration used: 5μg/ml) for 15 minutes before harvest. GFP+ and GFP-cells, as well as cells in G1 and G2 phases were sorted using BD Influx Sorter (BD Biosciences; see S8A Fig for typical gating plots). Total RNA was extracted from 50 000 cells following RNAqueous Total RNA isolation kit (Life technologies) instructions. cDNA was synthesized from 150ng of RNA using SuperScript IV (Life Technologies) according to manufacturer's instructions. Real time quantitative PCR were performed with the Prime Pro 48 real time qPCR system (Techne) using Absolute SYBR Capillary Mix (Thermo Fisher Scientific) and the primers below. The expression of each gene was normalised to that of *TBP*. Data representation and statistical analyses using *t*-test were performed in GraphPad Prism.

 *P03: LMO4-Fw: GGCACGTCCTGTTACACCAA*
 *P04: LMO4-Rev: CGCCCTCATGACGAGTTCAC*
 *P05: NGFR-Fw: GCCTGTACACACACAGCGG*
 *P06: NGFR-Rev: ACCACGTCGGAGAACGTCAC*
 *P07: PAX3-FOXO1-Fw: TCCAACCCCATGAACCCC*

*P08: PAX3/7-FOXO1-Rev: GCCATTTGGAAAACTGTGATCC*
*P09: PAX7-FOXO1-Fw: CAACCACATGAACCCGGTC*
*P10: PIPOX-Fw: ACCCGGATGATGCATGAGTG*
*P11: PIPOX-Rev: ACCCTCTGCCTCGACAGATT*
*P12: SERPINI1-Fw: AAAATGTAGCCGTGGCCAACTA*
*P13: SERPINI1-Rev: GGCAGCATCAAAATCCCTTGG*
*P14: SKP2-Fw: TTTGCCCTGCAGACTTTGCTA*
*P15: SKP2-Rev: TCTCTGACACATGCGCAACA*
*P16: TBP-Fw: CACGAACCACGGCACTGATT*
*P17: TBP-Rev: TTTTCTTGCTGCCAGTCTGGAC*
*P18: TFAP2α-Fw: GAGGTCCCGCATGTAGAA*
*P19: TFAP2α-Rev: CCGAAGAGGTTGTCCTTGT*

## Supporting information

**S1 Fig. (a) Schematics showing the homogenous and diverging structural protein domains of PAX3-FOXO1 and PAX7-FOXO1.** The two proteins share 77% of their amino-acid sequence. The position of the domains displaying divergence in their amino-acid sequence are highlighted by colour coded stars; green stars: <5% of divergence; orange stars: <20% of divergence; red stars: >20% of divergence. HD: homeodomain; OP: octapeptide; PrD: Paired DNA binding domain. **(b-e) Levels of expression of PAX3-FOXO1 and PAX7-FOXO1 in DOX inducible HFF lines. (b)** Stable HFF lines were generated by introducing a DOX-inducible *2xFLAG-PAX3-FOXO1* or *2xFLAG-PAX7-FOXO1* cassette between exons 1 and 2 of the *PPP1R12C* gene at the *AAVS1* locus. **(c)(i)** Western blots using FOXO1 and βACTIN antibodies on protein extracts from three independent HFF cell lines (1 to 3) expressing either PAX3-FOXO1 (P3F) or PAX7-FOXO1 (P7F), untreated (Ø) and treated with DOX for 48h (DOX). Arrows indicate PAX-FOXO1 band, black arrowheads point at FOXO1 band and the blue one at βACTIN. **(ii)** Relative PAX-FOXO1 levels to that of βACTIN quantified from western blots in (i). **(d)** DAPI staining and immuno-staining using antibodies recognizing GFP, PAX homeodomain (HD) and FOXO1 C-terminal domain on the indicated HFF cell lines treated for 48h with DOX (bottom panels) or without (top panels). Only upon DOX treatment a specific nuclear signal is revealed by the three antibodies in the 3 independent lines expressing PAX3-FOXO1 or PAX7-FOXO1 (#1 to 3). Scale bar: 10μm. **(e)** Quantification of the levels of PAX-HD signal in GFP+ cells showing an equivalent dispersion of signal levels in PAX3-FOXO1 (P3F) and PAX7-FOXO1 (P7F) expressing cell lines (dots: value for a cell in arbitrary units (AU); bars: mean ± s.d.; ns: ANOVA *p*-value higher than 0.05).
(TIF)

**S2 Fig. Reproducibility and validation of PAX3-FOXO1 and PAX7-FOXO1 genome occupancy assessment using CUT&Tag. (a)** Heatmaps of normalized CUT&Tag signals obtained using either an anti-FLAG or an anti-FOXO1 in control (Ctl) or 3 independent PAX3-FOXO1 (P3F-1 to 3) and PAX7-FOXO1 (P7F-1 to 3) expressing HFF 48h post-DOX treatment in clusters of *cis*-regulatory modules (CRMs) defined in Fig 1B. Peaks were arranged vertically following the order of signals obtained in P3F-1 sample (cf. arrow). **(b)** Distribution of the selected 6000 PAX-FOXOs bound CRMs in functional genomic regions (expressed as percentage). TSS: Transcription Start Site. **(c) i.** Logos of position weight matrix of DNA recognition motifs for FOXO1. **ii.** Enrichment for FOXO1 DNA recognition motif displayed in i in the CRMs belonging to the clusters defined in Fig 1B (bars: -log2(*p*-value)). **(d)** Venn diagrams showing the overlap between the 6000 PAX-FOXO1 bound CRMs identified in HFF and all possible PAX3-FOXO1 bound CRMs that have been identified in Rh4, Rh3, SCMC [1] **(i)** and

the overlap between the 6000 PAX-FOXO1 bound CRMs identified in HFF and the list of high confidence PAX3-FOXO1 bound CRMs defined by [1] **(ii)**. **(e)** IGV tracks showing normalized FLAG and H3K27ac CUT&Tag reads distribution in HFF expressing either PAX3-FOXO1 (P3F) or PAX7-FOXO1 (P7F) 48h post-DOX treatment at four previously identified PAX3-FOXO1 target loci in Rh4 RMS cells [1]. Scales in counts per million (CPM). The peaks of reads overlap greatly with genomic regions identified by ChIPSeq as bound by PAX3-FOXO1 in Rh4, highlighted in blue [1].
(TIF)

**S3 Fig. Chromatin status of PAX3-FOXO1 and PAX7-FOXO1 bound regions. (a)** IGV tracks showing normalized FLAG, H3K27ac, H3K27me3 and H3K9me3 CUT&Tag reads distribution in HFF expressing either PAX3-FOXO1 (P3F) or PAX7-FOXO1 (P7F) 48h post-DOX treatment. Scales in counts per million (CPM). They demonstrated the efficiency of the CUT&Tag approach to reveal heterochromatic regions. They also showed at the *TNNT2* and *KCNN3* loci a poor overlap between PAX-FOXO1 recruitment sites and peak signals for the deposition of the H3K27me3 and H3K9me3 marks. Conversely, they indicated that signals for PAX-FOXO1 binding and H3K27ac deposition coincide. **(b)** Correlogram displaying the spearman correlations between the genome-wide distributions of FLAG, H3K27ac, H3K27me3 and H3K9me3 CUT&Tag reads in control HFF (Ctl), P3F and P7F cells 48h post-DOX treatment. The correlation between PAX-FOXO1 binding and the H3K27ac deposition signals (blue rectangles) was stronger than that between PAX-FOXO1 binding signals and H3K9me3 or H3K27me3 deposition signals (black rectangles). **(c)** Average profiles (top panels) and heatmaps (bottom panels) of normalized CUT&Tag signals for the deposition of the indicated histone mark at the 6000 PAX-FOXO1s-bound CRMs defined in Fig 1B (PF$^+$ CRMs) and signal peaks for the indicated histone mark common to all cell lines analysed (common peaks). CUT&Tags were performed in Ctl HFF or expressing P3F and P7F 48h after DOX treatment. The ranking of the CRMs in the heatmaps follows the signal distribution in the P3F sample (cf arrow). Levels of deposition of all three histone marks in common CRMs were comparable between cell lines testifying the robustness in the normalization of CUT&Tag signals. The data presented in both panels also demonstrated that PAX-FOXO1 bound regions were mainly unmarked by the histone modifications, H3K27ac, H3K27me3 and H3K9me3, in control HFF. These regions remained poorly enriched for the deposition of the two heterochromatic histone marks upon PAX-FOXO1s expression. In contrast, a significant number of these regions displayed enhanced H3K27ac deposition signal in presence of PAX-FOXO1. In PAX3-FOXO1 samples less CRMs were mediated H3K27ac deposition than in PAX7-FOXO1 samples. **(d)** Average profiles (top panels) and heatmaps (bottom panels) of normalized H3K9me3 CUT&Tag signals at the 3 CRMs clusters defined in Fig 1B. These were obtained in Ctl HFF or expressing P3F and P7F 48h after DOX treatment. The ranking of the CRMs in the heatmaps, marked with a white arrow, follows the signals distribution in the P3-F sample. **(e)** Scattered plots showing the levels of P3F or P7F signals (Log10(FLAG CUT&Tag signal)) as the function of the levels of deposition of the indicated histone mark (Log10(CUT&Tag signal for the mark) at the 6000 single CRMs that can be bound by PAX-FOXO1 in HFF. None of the signals were correlated, but the binding signal for PAX7-FOXO1 with the H3K27ac deposition signals. **(f)** Principal component analysis (PCA) projection of the chromatin state assessed using H3K27ac enrichment onto the 6000 PAX-FOXO1 bound CRMs of the indicated cell lines 48h post-DOX treatment.
(TIF)

**S4 Fig. Comparative analysis of the transcriptome of cells expressing Pax3, Pax7, PAX3-FOXO1 or PAX7-FOXO1. (a) Data obtained on stable HFF lines described in S1 Fig. (ai)**

Hierarchical clustering based on Jensen-Shannon divergence of RNAseq based transcriptomes of cells described in (Fig 2A) and treated with DOX for 48h. Main segregation nodes have been numbered. The top segregation node separated all PAX-FOXO1 expressing cells from control cells, supporting a primary common mode of action of the two fusion factors. The secondary node clustered all PAX7-FOXO1 (P7F) samples apart from PAX3-FOXO1 (P3F) samples, arguing thus for each PAX-FOXO1 subtype imposing specific transcriptomic traits to cells. **(aii)** Correlogram displaying the Jensen-Shannon distances between the transcriptomes of control HFF (Ctl), P3F and P7F cells 48h post-DOX treatment. This shows that the transcriptome of P7F cells is more distant from that of Ctl cells than that of P3F cells. **(aiii)** Heatmap of 4 K-means clusters of genes showing significant variations in their expression between control, PAX3-FOXO1 (P3F) and PAX7-FOXO1 (P7F) expressing cells. Fold changes across samples are colour-coded in blue (minimum levels) to red (maximum levels). **(aiv)** IGV tracks showing normalized FLAG, H3K27ac, H3K27me3 and H3K9me3 CUT&Tag reads distribution in HFF expressing either PAX3-FOXO1 (P3F) or PAX7-FOXO1 (P7F) 48h post-DOX treatment. Scales in counts per million (CPM). The loci chosen were nearby genes encoding for either cell cycle (left column) or cell shape and motility (right column) regulators and that belong to the gene clusters defined in ii. The bar graphs display the levels of expression assayed by RNAseq for the gene present at these loci in Ctl, P3F and P7F samples. **(av)** Standardized and normalized expression levels of genes assayed by RNAseq and in the vicinity of CRMs bound by P3F and/or P7F and belonging to the three clusters of CRMs defined in Fig 1B (dots: expression of a given gene; bars: mean±s.e.m.; ANOVA $p$-value: ****: <0.0001). This indicates that genes nearby CRMs that are bound by P3F and P7F were induced by both PAX-FOXO1s, yet displayed higher expression levels in presence of P7F than in presence of P3F (left graph). Genes nearby CRMs preferentially bound by P3F were more induced by P3F than P7F (middle graph). Similarly, those nearby CRMs preferentially bound by P7F displayed much higher expression levels in presence of P7F than P3F (right graph). **(b) Data obtained on HFF 72 hours post-transfection** with plasmids expressing either Pax3 (P3), Pax7 (P7), PAX3-FOXO1 (P3F) and PAX7-FOXO1 (P7F) or the empty pCIG plasmid (Ctl) and the GFP bicistronically. **(bi)** DAPI staining and immuno-staining using antibodies recognizing GFP, PAX homeodomain (HD) and FOXO1 C-terminal domain (recognizing both PAX-FOXO1 and FOXO1 proteins) on the indicated transfected HFF. Scale bar: 10μm. **(bii)** Quantification of the levels of PAX-HD signal (top graph) or of FOXO1 (bottom graph) in GFP- cells (first bar) and GFP+ cells (the remaining bars) (dots: value for a cell in arbitrary units (AU); bars: mean ± s.d.; ns: t-test $p$-value higher than 0.05; t-test $p$-value: **<0.005; ****: <0.0001). The levels expression of all four TFs vary between cells as in stable lines; this dispersion is comparable between the PAX3-FOXO1, Pax7, and PAX7-FOXO1 samples, in Pax3 samples more cells display weaker levels of PAX-HD. **(biii)** Multidimensional scaling 2D visual projection of the transcriptomic state of three or six independent HFF batches. The transcriptome of P7 cells were the most diverging from that of control. P7F cells were further away from control cells than P3F cells and P3 cells were positioned between control and P3F cells.
(TIF)

**S5 Fig. Functional annotations of genes differentially expressed between PAX3-FOXO1 and PAX7-FOXO1 in patient biopsies and HFF. (a)** GSEA plots showing the enrichment for genes associated to RMS [2] in differentially expressed genes between control (Ctl) HFF and HFF expressing either PAX-FOXO1s (PF) or PAX3-FOXO1 (P3F) or PAX7-FOXO1 (P7F) or between P3F and P7F samples. NES: normalized enrichment score; FDR: false discovery rate. **(b, c)** Gene ontology enrichment for biological processes applied to genes enriched in PAX3-FOXO1 RMS biopsies compared to PAX7-FOXO1 FP-RMS biopsies (a) and conversely to

genes associated to PAX7-FOXO1 RMS biopsies (b). **(d, e)** Gene ontology enrichment for biological processes applied to genes enriched in PAX3-FOXO1 expressing HFF compared to PAX7-FOXO1 expressing HFF (c) and conversely to genes associated to PAX7-FOXO1 HFF (d). Related biological processes are highlighted in colour. Bars: -log2(adjusted *p*-value). (TIF)

**S6 Fig. Transcriptional dynamics of genes coding for cell-architecture and cell cycle regulators in HFF upon PAX3-FOXO1 and PAX7-FOXO1 exposure.** Heatmaps of K-means clustered genes differentially expressed between control HFF, PAX3-FOXO1 and PAX7-FOXO1 stable lines 48h post-DOX treatments. In **(a)** are clustered genes selected based on the function of the protein they encoded in regulating cell morphology, cell-cell, cell-matrix adhesions, cell motility, while in **(b)** are grouped genes coding from known cell cycle regulators. In presence of either PAX-FOXO1s, core cell cycle genes, such as checkpoints (*CHEK1/2*) and Aurora A (*AURKA*) kinases, the cyclins (*CCNA2*, *B1* and *B2*) and their dependent kinases (CDK1/2) or the cell-cycle TF MYC are downregulated. Conversely, anti-proliferating genes, including *BTG1*, *CDKL5*, *CCNG2* or *SIRT6* [3–6] are up regulated by both PAX-FOXO1s. PAX7-FOXO1 signature harboured actors of the G2 to M and M to G1 transitions, including members of the anaphase promoting complex (*ANAPC2*, *CDC27*) and of the chromosomal passenger complex (*AURKC*), and modulators of these transitions (*CDC14A*, *SKP2*) [7–9]. Conversely, genes more induced by PAX3-FOXO1 were enriched for negative regulators of the G1 to S transition, such as the cyclin inhibitor *CDKN1A* (p21$^{Cip1}$) or *JUNB* [10,11]. Fold changes across P3F and P7F expressing and control HFF are colour coded in blue (lower level) and in red (higher level). Names of genes have been coloured coded as a function of gene families. (TIF)

**S7 Fig. PAX-FOXO1 subtype specific cell morphology and cycle phenotypes are penetrant across independent cell lines.** Immunodetection against the indicated proteins, phalloidin staining of F-actin filaments and DAPI labelling in control or two independent PAX3-FOXO1 (P3F-1, 2) and PAX7-FOXO1 (P7F-1, 2) cell lines 48h post-DOX treatment. Arrowheads indicate GFP positive cells showing abnormally low levels of pRB1 or Ki-67 and arrows point at GFP positive cells in which these two proteins can still be detected. All the cells in panel xvii to xx are GFP positive (not shown). Scale bars: 15μm. (TIF)

**S8 Fig. The transactivation potential differential between PAX3-FOXO1 and PAX7-FOXO1 occurs regardless of cell cycle phase. (a)** Schematics showing that cells were treated for 48h with DOX and for 15 minutes with Hoechst before preparation for FACS sorts **(i)**. Typical FACS plots illustrating the segregation of GFP$^+$ (PAX-FOXO1$^+$) from GFP$^-$ cells **(ii;** y axis: GFP signal levels; x axis: Hoechst signal intensity) and the selection windows of cells in the G1 (blue) and G2 (pink) phases **(iii;** y axis: cell count; x axis: Hoechst signal intensity). **(b, d)** Levels of expression of indicated genes in G1 or G2 phases and in presence or not of PAX3-FOXO1 (P3F) and PAX7-FOXO1 (P7F) (circles: values obtained for independent cell lines, bars: mean ± s.e.m.). This shows that the expression levels of PAX3-FOXO1 and PAX7-FOXO1 induced by DOX from the *AAVS1* locus are not submitted to variation throughout the cell cycle (**b**). For all PAX-FOXO1 targets analysed, their expression levels varied between G1 and G2 phases. In contrast, these targets are in both phases induced by PAX-FOXO1 and their specific sensitivity to PAX3-FOXO1 or PAX7-FOXO1 remains the same. (**d**). Thus, PAX3-FOXO1 induced *NGFR* and *SERPINI1* to a greater extent than PAX7-FOXO1 regardless of the cycle phase. Similarly, *SKP2* and *PIPOX* showed higher expression levels in the presence

of PAX7-FOXO1 than PAX3-FOXO1 in G1 and G2. **(c)** Graph representing the ratio of the expression levels of the indicated genes in PAX3-FOXO1 cells to those in PAX7-FOXO1 cells in the G1 and G2 phases of the cell cycle, illustrating that the differential response of the cells to a given type of PAX-FOXO1 is not affected by the phase of the cell cycle (bars: log2 of the average over three independent cell lines).
(TIF)

**S1 Table. Position and normalized reads count distribution at the best 6000 PAX-FOXO1s bound *cis*-regulatory modules (CRMs) after CUT&Tag using FLAG, FOXO1, H3K27ac, H3K9me3 and H3K27me3 antibodies in control, PAX3-FOXO1, PAX7-FOXO1 cell lines (Genome Ref: hg19) and the closest gene.**
(XLSX)

**S2 Table. Nature and distribution of PAX-FOXO1 related motifs in PAX-FOXO1s bound *cis*-regulatory modules (CRMs).** Sheet 1: PWM for 21 DNA binding motifs previously identified for PAX3, PAX7 and PAX3-FOXO1 as well as bHLH and FOXO1. Sheet 2: Distribution of all 21 PWM across cluster 1, 2 and 3 defined in Fig 1B. The adjusted *p*-value was calculated by AME of the MEME-suite and correspond to "The optimal enrichment *p*-value of the motif according to the statistical test, adjusted for multiple tests using a Bonferroni correction. If the best *p*-value is *p* before adjustment, and the number of multiple tests is *n*, then the adjusted *p*-value is $1 - (1-p)^n$".
(XLSX)

**S3 Table. Differential analyses of PAX3-FOXO1 and PAX7-FOXO1 expressing HFF transcriptomes.** Sheet 1: Stable HFF lines. Genes RPKM for all samples; results of the pairwise differential analysis performed using Cufflinks. Genes with a *q*-value <0.05 after Benjamini-Hochberg correction for multiple-testing were considered to be significant; Cluster (A, B, C, D defined in Fig 2A); Functional association with cell morphology; Association with cell cycle regulation. Sheet 2: Transient transfected HFF. Genes RPKM for all samples; results of the pairwise differential analysis performed using Cufflinks. Genes with a *q*-value <0.05 after Benjamini-Hochberg correction for multiple-testing were considered to be significant.
(XLSX)

**S4 Table. Differential analysis of PAX3-FOXO1 and PAX7-FOXO1 expressing biopsies transcriptome [12].** Genes mean expression in PAX3-FOXO1 or PAX7-FOXO1 biopsies. Results of paired *t*-test (P3F/P7F). Only genes with a *p*-value <0.05 are shown. Association with PAX3-FOXO1 or PAX7-FOXO1.
(XLSX)

**S5 Table. GO term-analyses of PAX3-FOXO1 and PAX7-FOXO1 associated transcriptomes in RMS biopsies and HFF stable lines.** The table displays GO term, adjusted *p*-value and genes associated for each GO. GO with a *p*-value<0.05 are colour highlighted. Sheet 1: Results for genes associated with PAX3-FOXO1 in RMS biopsies. Sheet 2: Results for genes associated with PAX7-FOXO1 in RMS biopsies. Sheet 3: Results for genes associated with PAX3-FOXO1 in HFF stable lines. Sheet 4: Results for genes associated with PAX7-FOXO1 in HFF stable lines.
(XLSX)

**S6 Table. Matrices of the western blots, immunostaining and q-RT-PCR data graphed in the manuscript Figures.**
(XLSX)

## Acknowledgments

We deeply thank the ImagoSeine core facility of Institut Jacques Monod, a member of France-BioImaging (ANR-10-INBS-04) and certified IBiSA. Notably, S. Many and N. Valentin for performing flow cytometry analyses and N. Moisan for training us on confocal imaging. We are grateful to L. Vinel for helping us during her undergraduate internship and V. Doye and S. Nedelec for critical inputs on our manuscript. We are thankful to people who have provided us with useful tools. We received *pAAVS1-PDi-CRISPRn* from B. Conklin and gRNAs from G. Church; and HFF from M-C. Geoffroy; the anti-53BP1 and anti-γH2AX antibodies from C. Boumendil and the anti-PAX3/7HD from N. Patel. We thank Bérengère Guichard for the preparation and purification of pA-Tn5 enzyme.

## Author Contributions

**Conceptualization:** Line Manceau, Pascale Gilardi-Hebenstreit, Vanessa Ribes.

**Formal analysis:** Line Manceau, Pascale Gilardi-Hebenstreit, Vanessa Ribes.

**Funding acquisition:** Maxim V. C. Greenberg.

**Investigation:** Line Manceau, Pascale Gilardi-Hebenstreit, Vanessa Ribes.

**Methodology:** Line Manceau, Julien Richard Albert, Pascale Gilardi-Hebenstreit, Vanessa Ribes.

**Project administration:** Vanessa Ribes.

**Resources:** Pier-Luigi Lollini, Maxim V. C. Greenberg, Vanessa Ribes.

**Software:** Julien Richard Albert.

**Supervision:** Pascale Gilardi-Hebenstreit, Vanessa Ribes.

**Validation:** Line Manceau, Pascale Gilardi-Hebenstreit, Vanessa Ribes.

**Visualization:** Line Manceau, Julien Richard Albert, Vanessa Ribes.

**Writing – original draft:** Line Manceau, Vanessa Ribes.

**Writing – review & editing:** Line Manceau, Julien Richard Albert, Maxim V. C. Greenberg, Pascale Gilardi-Hebenstreit.

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
