## [Decision Letter · Decision Letter 0]

5 Oct 2021

Dear Dr Ribes,

Thank you very much for submitting your Research Article entitled 'Divergent transcriptional and transforming properties of PAX3-FOXO1 and PAX7-FOXO1 paralogs' to PLOS Genetics.

The manuscript was fully evaluated at the editorial level and by independent peer reviewers. The reviewers appreciated the attention to an important problem, but raised some substantial concerns about the current manuscript. Based on the reviews, we will not be able to accept this version of the manuscript, but we would be willing to review a much-revised version. We cannot, of course, promise publication at that time.

If you decide to revise the manuscript for further consideration at PLOS Genetics, please aim to resubmit within the next 60 days, unless it will take extra time to address the concerns of the reviewers, in which case we would appreciate an expected resubmission date by email to plosgenetics@plos.org.

[LINK]

We are sorry that we cannot be more positive about your manuscript at this stage. Please do not hesitate to contact us if you have any concerns or questions.

Yours sincerely,

Mark E. Hatley

Guest Editor

PLOS Genetics

David Kwiatkowski

Section Editor: Cancer Genetics

PLOS Genetics

Reviewer's Responses to Questions

**Comments to the Authors:**

Reviewer #1: minor comments:

(1) "Both PAX-FOXO1s result in related cell transformation in animal models, but both mutations are associated with distinct pathological manifestations in patients".

Of note, PAX-translocations are not "mutations" but rather alterations. Defining the chromosomal translocations as mutations is incorrect and potentially confusing.

(2) "the identified pronounced deleterious effects of PAX7-FOXO1 provide an explanation for the low frequency of the translocation generating this factor in patients with rhabdomyosarcoma"

In patients, PAX7 fusions often have better clinical outcomes than PAX3 fusions. The statement, as written might be confusing to readers, suggesting that PAX7-fusions have worse clinical outcomes.

major comments:

(1) It is of note that PAX3 and PAX7 fusions are expressed as distinct splice forms. This has been known for more than 15 years (PMID: 15688409). In the literature on this topic, PAX7 fusions seem to be co-expressed with about four distinct splice variants, while often the subject of gene amplification events. Merely expressing the fusions in differentiated cells will not recapitulate the clinical/physiological features of these genes as they are expressed in human cancer.

(2) "Finally, each of the PAX-FOXO1s could have its own transcriptional activity. Supporting this idea, PAX3-FOXO1 and PAX7-FOXO1 RMS differ in their DNA methylation profiles and transcriptome".

What is the relationship between chromatin binding and transcription for these fusions? As written it seems that the authors define DNA-methylation as a bridge between molecular recognition of DNA and altered transcription. However, DNA methylation occurs on very slow timescales, while transcriptional activity occurs on very rapid timescales.

(3) "PAX3-FOXO1 binding to non-coding cis-regulatory modules (CRMs) is mediated by its PrD alone or in combination with its HD, but rarely by its HD alone [15,16]"

The conclusion that the FOXO domains do not play a role in chromatin recognition seems to be based on incomplete evidence. What is the evidence that both parts of the fusion do not regulate DNA recognition? The experimental design is potentially problematic, "Human foreskin fibroblasts (HFF) were engineered to express a copy of a FLAG-tagged version of these TFs in a doxycycline (DOX)-inducible manner from the AAVS1 safeguard locus (Figure S1b)." By introducing a tag onto the fusion oncoprotein, the sterics of chromatin recognition will likely be altered. Thus, conclusions from this type of tag-based system are specific for the FLAG-fusions, more so than the endogenous PAX3/7-FOXO fusions found in patients.

(4) "Using these cell lines, we first mapped the genomic targets of PAX3-FOXO1 and PAX7- FOXO1 using Cleavage Under Targets and Tagmentation (CUT&Tag) experiments [22] (Figure 92 1; S2)."

The molecular basis of Cut&Tag relies on an "input" sample that is essentially ATAC-seq, or open chromatin. Then, the “IP” relies on the pre-condition of chromatin that is already accessible in order to isolate epitope specific DNA-protein binding events. The system is based on selecting accessible DNA-binding sites for the fusion oncoproteins. Recent evidence in the field suggests that repressive or quiescent chromatin might serve as key targets for these fusions, and this information cannot be captured with this experimental system.

This experimental selection for accessible chromatin is further underscored with the "Recruitment of PAX3-FOXO1 and PAX7-FOXO1 to the genome was generally correlated with de novo deposition of H3K27ac (Figure 1a,b,civ-vi)" statement. The correlation of high levels of K27ac (accessible chromatin, overlapping with ATAC signals) and PAX fusion binding is a product of the experimental design, and likely not the true biology and chromatin recognition of the system.

(5) Throughout the text, it is suggested that the HD and PD domains (PAX domains) of the fusions regulate DNA binding. However, then it is stated that "the induction of specific gene cohorts by one of the PAX-FOXO1 chimeras is a property that emanates from its PAX moiety, whereas the induction of common genes is property of the fusion protein."

It is difficult to “rule out” any contributions form the FOXO domain in chromatin recognition of unique sites, to result in distinct transcriptional outcomes. Moreover, it is not discussed in the text whether there are apparent gene-repressive roles for the fusions as well. This should be stated, and elaborated upon. Presenting or focusing only on gene-activating roles is inconsistent with key studies in the field, and potentially driven from the experimental techniques used in this study.

This should be included, but the limitations of the CUT&Tag system (see above) might make the argument for gene repression more difficult to describe with the proposed experimental model.

(6) The cell cycle features of the two oncoproteins are meritorious of further mechanistic examination. The preference for G1/S vs G2/M phases for the PX3 and PAX7-fusions should be discussed also in the context of previous established studies on "checkpoint adaptation" for PAX3FOXO1 (PMID: 24453992).

Altogether, the study is promising, but the methodological limitations especially for the genome-wide binding studies make it challenging to evaluate the conclusions in the context of the approaches and data presented.

Connections with cell cycle specific expression (PMID: 24453992), gene amplification for fusions (PMID: 15688409), and sampling chromatin outside of accessible regions would strengthen this study.

Reviewer #2: 1. The authors show sites of PAX3- and PAX7-FOXO1 binding, and related H3K27ac that tracks along with clusters 1/2/3 that have P3F/P7F. At these same sites, what is the signal from publically available H3K27ac (cell lines and tumors) or PAX3-FOXO1 signal (in cell lines)? Note: H3K27ac data from PAX7-FOXO1 primary tumor has been deposited, but not analyzed for the kinds of comparisons done between P7F and P3F in this paper. It would be interesting here if the authors directly compared H3K27ac from a PAX7-FOXO1 tumor (https://www.ncbi.nlm.nih.gov/geo/query/acc.cgi?acc=GSM2214089 ) to H3K27ac from a PAX3-FOXO1 tumor (https://www.ncbi.nlm.nih.gov/geo/query/acc.cgi?acc=GSM4058221 ). PAX3-NCOA1 ChIP-seq is in the iScience paper (https://pubmed.ncbi.nlm.nih.gov/32416589/) but also hasn’t been analyzed genome wide (https://www.ncbi.nlm.nih.gov/geo/query/acc.cgi?acc=GSM4058219 ).

2. Two of the most iconic locations of PAX3-FOXO1 binding are at the MYOD1 and FGFR4 super enhancers. Are either of those recapitulated in these CUTandTag experiments?

3. Figure 1 panel C would benefit from showing the PrD and HD domains (showing the actual motifs) and using, once, the full name next to the abbreviations PrD and HD.

4. Toward the middle of the manuscript, associated with Figure 2, the authors describe differences in transcriptional output downstream of PAX3-FOXO1/PAX7-FOXO1 activity. In the cell line models, no gene ontology is reported, yet for the primary tumor analysis of public microarray data they do describe sets of gene ontology (ie, cell cycle, cell migration, and metabolism for PAX3-FOXO1 selectively upregulated genes, and embryonic lineage differentiation as well as genes involved in cytoskeletal remodelling for PAX7-FOXO1). The authors need to (a) perform gene ontology from the cell lines models and see if it agrees with their existing analysis, and (b) make two “gene sets” of these 988 and 856 genes (Figure 2d) and use them to perform GSEA on the log2 fold change of expression from their experiments in Figure 2a/c.

5. The paper is all about “Divergent transcriptional and transforming properties of PAX3-FOXO1 and PAX7-FOXO1” and generated new CUTandTag data and new RNA-seq data from a model system in fibroblasts. Based on this, it would be expected that at least 2 types of example gene loci (genome browser shots) would be shown: 1 PAX3-FOXO1 bound and selectively upregulated gene, 1 PAX7-FOXO1 bound and selectively upregulated gene, and 1 gene bound by and upregulated by either P3F/P7F. This kind of “zoom in” to illustrate the transcriptional divergence shown in Figure 2 are warranted.

6. The authors would do well to perform GSEA on the RNA-seq data in cells +/-PAX3-FOXO1 and PAX7-FOXO1, specifically using some of the MolSigDB genesets known for PAX3-FOXO1”

https://www.gsea-msigdb.org/gsea/msigdb/cards/EBAUER_MYOGENIC_TARGETS_OF_PAX3_FOXO1_FUSION

https://www.gsea-msigdb.org/gsea/msigdb/geneset_page.jsp?geneSetName=GRYDER_PAX3FOXO1_ENHANCERS_KO_DOWN

https://www.gsea-msigdb.org/gsea/msigdb/geneset_page.jsp?geneSetName=DAVICIONI_PAX_FOXO1_SIGNATURE_IN_ARMS_UP

7. “Deeper alterations of cell architecture and cell cycle induced by PAX7-FOXO1 than by PAX3-FOXO1” is missing a verb, such as “are induced”. I would recommend active voice here anyway, so “PAX7-FOXO1 induces deeper alterations in cell architecture and cell cycle than PAX3-FOXO1”.

Reviewer #3: In their manuscript, Manceau et al. present a deep analysis of the effects of PAX3-FOXO1 and PAX7-FOXO1 on the function of normal human fibroblasts. The goals of the study are clear – to describe the distinct and common functions of these chimaeric fusion genes on the transcriptome, epigenome and cell biology of untransformed cells. This is conducted through the generation of models of epitope-tagged PAX3-FOXO1 and PAX7-FOXO1 in HFF cells, followed by study of genome-wide localization of these fusion proteins by CUT&TAG, then analysis of transcriptomics including some fusion with their epigenomic data, and finally, a study of the cell biology of this de novo system. In general, the data is internally consistent, well presented and intriguing. I do, however, have three major concerns and several minor concerns that should be addressed prior to the manuscript being suitable for publication. The goals of my comments are to strengthen this manuscript and ensure experimental rigor.

Major concerns:

1. In their CUT&TAG analysis, the authors present data to demonstrate that levels of the H3K27ac mark is higher in PAX7-FOXO1 expressing samples, compared with PAX3-FOXO1 expressing samples. In a review of their text, legends and methods section, I am unable to determine if these samples were spiked-in with an exogenous control of some sort. E.coli DNA would be standard for CUT&TAG analysis, and this would be used for normalization of the data. In the absence of this important control, the statement that PAX7-FOXO1 binding is higher than PAX3-FOXO1 binding is difficult to draw and challenging to intepret. Was this control performed? If not, how were the data normalized?

2. In general, I find several examples of data that are over-interpreted. The manuscript is of interest by itself, and does not require the data to be “sold.”I am listing several examples below, which should be corrected, however, there may be further examples.

a. Line 127: “Therefore, both factors induce active chromatin signatures in healthy cells.” The data here argues that PAX/FOXO fusion proteins are found at sites marked by H3K27ac. Presence of the factor is not the same as chromatin remodeling and establishment of H3K27ac – essentially, induction of the chromatin signature. Recent work by Ben Stanton’s group (iScience 2021) seems to suggest PAX3-FOXO1 is a pioneer factor, which would argue this point, but the data as presented in the paper does not substantiate the claim. Rather, it reflects a correlation. Also, the point of “healthy” cells is unclear. Do the authors mean untransformed cells?

b. Line 153: “Thus, PAX7-FOXO1 also emerges as a more potent transactivator than PAX3-FOXO1 in fully transformed contexts.” This claim is based on the finding that H3K27ac is higher in one PAX7-FOXO1+ cell line (RMZ-RC2), and not in another (CW9019) which has lower levels of PAX7-FOXO1. Most importantly, co-binding is not evidence of gene transactivation, which requires evidence of gene upregulation (not discussed until later in the manuscript). This statement should be softened appropriately.

c. Line 231: “This would certainly lead to two discrete modes of tissue invasions.” This statement immediately follows a segment describing changes in nuclear architecture. I believe the authors have demonstrated intriguing data on actin stress fiber formation. This may be what the authors are referring to, though even actin stress fiber formation is not causal for tissue invasion. If the authors wish to include this statement in the manuscript, it would be important to substantiate that these cells are, in fact, invasive, and that PAX3-FOXO1 and PAX7-FOXO1 positive cells have different methods of tissue invasion. In the absence of this data, I would strongly suggest this statement be removed and the text be clarified.

d. Line 252: “PAX7-FOXO1 was more prone to cause genomic instabilities than PAX3-FOXO1.” This statement is over-interpreted. The authors demonstrate increased gamma-H2AX foci, which is a marker of DNA double strand breaks. Is it possible that PAX7-FOXO1 causes a preferential defect of DNA repair mechanisms? Fundamentally, increased double strand breaks are not the same thing as whole genome instability. This statement should be softened/clarified.

3. The authors present an intriguing analysis of PAX3-FOXO1 and PAX7-FOXO1 signature genes in Figure S4a. Does this signature correlate to the signature induced in the HFF cells? I can imagine that it may not, due to expression of the transgenes in untransformed cells, though if there are correlations, then this would be interesting to comment on. If there are not, the authors should explain why not in the text.

Minor Concerns:

1. Many contractions are not defined in the manuscript – notable “CRMs,” “PrD”, “HD,” “bHLH,” “TF” and so on. Careful attention to the writing of the manuscript should be paid in order to ensure that abbreviations are defined and the manuscript is clear.

2. As a control in Figure 1d, the authors should present H3K27ac data in normal HFF cells.

3. The western in Figure S1 appears to demonstrate that FOXO1 expression is increased by doxycycline – is this true/recurrent? Is it due to the transgene or doxycycline? The authors should explain this.

4. H3K27ac is a chromatin mark. The term “occupancy” refers to transcription factors or other proteins that bind to DNA.

5. The authors should include a statement that other translocations are found in the disease (i.e. NCOA2, VGLL2).

6. Some evidence of expression of murine Pax3 or Pax7 should be presented – either qPCR or ideally a western blot – to support Fig 2c. Further, the authors refer only to expression in “cells” – do they mean HFF cells?

7. Are RDabl cells the same as RD?

8. Are there effects from integrating into the PPP1R12C locus?

**Have all data underlying the figures and results presented in the manuscript been provided?**

Reviewer #1: Yes

Reviewer #2: Yes

Reviewer #3: Yes

PLOS authors have the option to publish the peer review history of their article (what does this mean?). If published, this will include your full peer review and any attached files.

Reviewer #1: No

Reviewer #2: **Yes: **Berkley E. Gryder

Reviewer #3: No

---

## [Decision Letter · Decision Letter 1]

30 Mar 2022

Dear Dr Ribes,

Thank you very much for submitting your Research Article entitled 'Divergent transcriptional and transforming properties of PAX3-FOXO1 and PAX7-FOXO1 paralogs' to PLOS Genetics.

The manuscript was fully evaluated at the editorial level and by independent peer reviewers. The reviewers appreciated the attention to an important topic but identified some concerns that we ask you address in a revised manuscript

We therefore ask you to modify the manuscript according to the review recommendations. Your revisions should address the specific points made by each reviewer.

[LINK]

Yours sincerely,

Mark E. Hatley

Guest Editor

PLOS Genetics

David Kwiatkowski

Section Editor: Cancer Genetics

PLOS Genetics

Dear Dr. Ribes,

Thank you for your patience during the review process. In general, the reviewers were satisfied with your re-submitted manuscript and how it addressed the original review. Reviewers 3 and 4 have noted things that should be addressed either with further explanation or removal from the text. No new experiments were requested (or necessary), but please consider the reviewers' comments when revising your manuscript.

Best regards,

Mark Hatley

Reviewer's Responses to Questions

**Comments to the Authors:**

Reviewer #2: The authors have nicely addressed all my concerns, and the paper is improved greatly. I'm happy to recommend publication.

Reviewer #3: Manceau et al present a revised manuscript, defining the effects of PAX3-FOXO1 and PAX7-FOXO1 on the transcriptome and epigenome of fibroblasts. It is clear they have done much work in this revision, and the manuscript is improved as a result. Despite this, I still find issues that must be addressed for this manuscript to be acceptable for publication. I do not think further experiments need to be done - rather, I think that the authors need to carefully edit this manuscript for clarity and to ensure their statements reflect their findings.

1. The authors broadly overstate their findings in several places. The most concerning of these overstatements is found in the abstract, summary, results and discussion, where the authors claim that PAX7-FOXO1 has "higher transactivation potential" than PAX3-FOXO1. Their data as presented does not support this very broad claim. At no point do the authors demonstrate that the whole transcriptomic effects of PAX7-FOXO1 expression are greater than those of PAX3-FOXO1. This could be achieved by a violin plot of all expressed genes in these different scenarios. Alternatively, the authors could focus on genes bound by both PAX3-FOXO1 and PAX7-FOXO1 and examine the actual gene expression values in their RNAseq data. Is the expression higher for PAX7-FOXO1 than PAX3-FOXO1 at loci bound by both these proteins? The authors attempt an analysis like this in Fig 2C, however, this is a row-normalized heatmap, and a scatterplot or bar plot with an appropriate statistical test would actually prove this.

Regardless, this point is actually not exceptionally important. PAX3-FOXO1 and PAX7-FOXO1 bind to similar and distinct regions of the genome, both function as pioneer factors, and cause distinct functional outcomes. Claims of higher or lower transactivation, in the absence of showing actual gene transactivation, are really unnecessary.

To this end, I strongly recommend the authors simply remove this concept of "higher" or "lower" transactivation from their manuscript. Their data does not support this statement, and it doesn't really add much to their paper.

Other examples of broad overstatements:

Line 168-175: Here, the authors describe that H3K27ac is deposited at CRMs higher at P7F loci than P3F loci. The authors go on to discuss that this means that there is higher transactivation potential for P7F. H3K27ac is a histone mark, associated with promoters and enhancers. It correlates with expressed genes, but is also found at poised loci which are not expressed. It is an overreach to claim that more H3k27ac means more gene transactivation, without providing further evidence.

Line 302-304: Here, the authors describe that they will test whether PAX-FOXO cell cycle deregulation is linked to DNA damage. They then go on to show that DNA ds breaks are elevated in PAX7-FOXO1 expressing cells, compared with PAX3-FOXO1, and then claim that these means more genome instability. This section needs to be re-written. The authors have shown correlative data to suggest that PAX7-FOXO1 expression is associated with increased DNA ds breaks, and have not linked this to cell cycle control at all. Further, the presence of DNA ds breaks is not the same thing as genome instability.

2. Minor text edits:

Line 123-124: The authors state that they identified 6000 CRMs with PAX3-FOXO1 and/or PAX7-FOXO1 binding signals. In the methods, however, they state that they took the "top 6000" binding signals. Which is true?

Comments on RMS cell lines should be taken out of the text, now that they are not used in this manuscript anymore.

The discussion of EWS-FLI and drawing parallels to PAX3-FOXO1 and PAX7-FOXO1 in the discussion is confusing.

Reviewer #4: Manceau et al directly compares the genome binding of PAX3-FOXO1 and PAX7-FOXO1, associated histone marks, and transcriptional consequences in the context of human fibroblasts. This has been a longstanding question in the rhabdomyosarcoma field, why PAX7-FOXO1 has different clinical outcomes than PAX3-FOXO1 positive tumors. This is a challenging question to address given the lack of animal models and high toxicity of the fusion-oncogenes in cell culture systems. This study represents a first step to understand PAX3/7-FOXO1 divergent activities and ties their genomic data to phenotypic consequences detailed in Figure 3 related to cellular morphology, cell cycle arrest, and DNA integrity. The authors have addressed the reviewer comments satisfactorily and have presented new data that has improved the overall study. Additional clarification on whether these fibroblasts have the capacity to transform would help frame this study and claims in the context of the field (see below).

Major Comments

1. Additional explanation on the rationale for choosing fibroblasts to express the PAX3/7-FOXO1 fusions.

Elaborate on why fibroblasts were chosen for this study instead of myoblasts. Do these PAX3/7-FOXO1 expressing fibroblasts have the capacity to transform, and if not, how would you interpret these findings as being related to the disease? This could be included as part of the discussion.

Minor Comments

1) There is no limit to PLOS Genetics manuscript length and the methods can be included in the main text.

2) Figure 2A legend does not describe the presented data.

3) Figure 3A is a red/green overlay and should be modified so it is accessible to colorblind individuals.

**Have all data underlying the figures and results presented in the manuscript been provided?**

Reviewer #2: Yes

Reviewer #3: Yes

Reviewer #4: Yes

PLOS authors have the option to publish the peer review history of their article (what does this mean?). If published, this will include your full peer review and any attached files.

Reviewer #2: **Yes: **Berkley E Gryder

Reviewer #3: No

Reviewer #4: No

---

## [Editor Report · Decision Letter 2]

25 Apr 2022

Dear Dr Ribes,

We are pleased to inform you that your manuscript entitled "Divergent transcriptional and transforming properties of PAX3-FOXO1 and PAX7-FOXO1 paralogs" has been editorially accepted for publication in PLOS Genetics. Congratulations!

Yours sincerely,

Mark E. Hatley

Guest Editor

PLOS Genetics

David Kwiatkowski

Section Editor: Cancer Genetics

PLOS Genetics

Comments from the reviewers (if applicable):

**Data Deposition**

http://datadryad.org/submit?journalID=pgenetics&manu=PGENETICS-D-21-01090R2

**Press Queries**

---

## [Editor Report · Acceptance letter]

18 May 2022

PGENETICS-D-21-01090R2 

Divergent transcriptional and transforming properties of PAX3-FOXO1 and PAX7-FOXO1 paralogs 

Dear Dr Ribes, 

We are pleased to inform you that your manuscript entitled "Divergent transcriptional and transforming properties of PAX3-FOXO1 and PAX7-FOXO1 paralogs" has been formally accepted for publication in PLOS Genetics! Your manuscript is now with our production department and you will be notified of the publication date in due course.

With kind regards,

Anita Estes

PLOS Genetics

On behalf of:
